# Deep Generative Models for the Discovery of Antiviral Peptides Targeting Dengue Virus: A Systematic Review

**DOI:** 10.3390/ijms26136159

**Published:** 2025-06-26

**Authors:** Huynh Anh Duy, Tarapong Srisongkram

**Affiliations:** 1Graduate School in the Program of Research and Development in Pharmaceuticals, Faculty of Pharmaceutical Sciences, Khon Kaen University, Khon Kaen 40002, Thailand; huynhanhduy.h@kkumail.com; 2Department of Health Sciences, College of Natural Sciences, Can Tho University, Can Tho 900000, Vietnam; 3Division of Pharmaceutical Chemistry, Faculty of Pharmaceutical Sciences, Khon Kaen University, Khon Kaen 40002, Thailand

**Keywords:** Dengue virus, antiviral peptides, deep generative models, variational autoencoders, generative adversarial networks

## Abstract

Dengue virus (DENV) remains a critical global health challenge, with no approved antiviral treatments currently available. The growing prevalence of DENV infections highlights the urgent need for effective therapeutics. Antiviral peptides (AVPs) have gained significant attention due to their potential to inhibit viral replication. However, traditional drug discovery methods are often time-consuming and resource-intensive. Advances in artificial intelligence, particularly deep generative models (DGMs), offer a promising approach to accelerating AVP discovery. This report provides a comprehensive assessment of the role of DGMs in identifying novel AVPs for DENV. It presents an extensive survey of existing antimicrobial and AVP datasets, peptide sequence feature representations, and the integration of DGMs into computational peptide design. Additionally, in vitro and in silico screening data from previous studies highlight the therapeutic potential of AVPs against DENV. Variational autoencoders and generative adversarial networks have been extensively documented in the literature for their applications in AVP generation. These models have demonstrated a remarkable capacity to generate diverse and structurally viable compounds, significantly expanding the repertoire of potential antiviral candidates. Additionally, this report assesses both the strengths and limitations of DGMs, providing valuable insights for guiding future research directions. As a data-driven and scalable framework, DGMs offer a promising avenue for the rational design of potent AVPs targeting DENV and other emerging viral pathogens, contributing to the advancement of next-generation therapeutic strategies.

## 1. Introduction

Viral infections represent a critical global health challenge due to their rapid transmission and substantial human and economic consequences. A 2020 report by the Institute of Labor Economics estimated the economic impact of eight major diseases, including dengue, at USD 8 trillion, with over 156 million life-years lost in 2016 [1]. Notably, seven of these diseases are viral in origin, underscoring the pressing need for effective strategies to mitigate their burden. Dengue fever continues to pose a significant global health threat, disproportionately affecting low- and middle-income countries due to systemic issues such as inadequate healthcare infrastructure, poor sanitation, and limited access to preventive measures. An analysis of global data from 1990 to 2021 reveals a concerning rise in dengue-related mortality, nearly doubling from 14,315 deaths in 1990 to 29,076 deaths in 2021, accompanied by sharp increases in disease incidence and prevalence [2]. These trends underscore the critical need for targeted interventions to address this escalating public health crisis.

The prevention and management of viral infections remain a formidable challenge due to their efficient replication mechanisms, diverse transmission routes, and high genetic variability. In the case of the DENV, despite decades of extensive research, no antiviral drugs have yet been approved for clinical use, leaving treatment confined to supportive measures such as fluid replacement and analgesics [3]. Drug development efforts have faced setbacks, with many antiviral candidates failing to progress due to suboptimal physicochemical and pharmacokinetic properties. Notably, candidates like chloroquine, prednisolone, lovastatin, and celgosivir underwent clinical trials but failed to achieve sufficient viremia reduction to provide meaningful therapeutic benefits [4]. This underscores the urgent need for innovative strategies in dengue antiviral development.

The escalating health threat posed by dengue underscores the urgent need for innovative antiviral therapies. AVPs, a specialized class of antimicrobial peptides (AMPs), hold considerable potential for combating viral diseases [5,6]. These peptides interfere with the viral replication cycle through mechanisms such as disruption of viral membranes, inhibition of viral entry into host cells, and prevention of viral budding [7,8]. Notably, multiple AVPs have been patented and approved by the U.S. Food and Drug Administration (FDA) for the treatment of viruses such as HIV, influenza, and hepatitis, showcasing their efficacy as potent therapeutic agents [9]. However, despite their therapeutic promise, the availability of comprehensive data on AVPs remains significantly limited. Extensive repositories of natural and synthetic AMPs, including APD3, DRAMP 2.0, CAMP-R3, dbAMP, and ADAM, are available; however, only a small proportion of these entries specifically annotate AVPs. Furthermore, databases exclusively dedicated to AVPs, such as AVPdb and HIPdb, remain scarce. Collectively, these datasets encompass over 29,000 unique peptides, yet AVPs constitute merely 1–7% of the total entries [8]. Expanding specialized datasets for AVPs is imperative to advancing antiviral research and fostering the development of effective therapeutics, particularly for diseases like DENV.

In recent years, the rapid advancement of deep learning methodologies has revolutionized peptide science, enabling significant progress in peptide generation and identification [10,11]. Building upon recent advancements, this report offers a comprehensive analysis of the application of cutting-edge deep learning-based generative models for the design and development of AVPs, with a specific emphasis on targeting the DENV. It explores advanced computational methodologies, including generative adversarial networks (GANs) and variational autoencoders (VAEs), which have emerged as pivotal tools for peptide discovery, particularly in the identification of AVPs. These approaches are regarded as promising strategies for guiding future experimental investigations in related fields.

## 2. Methodology

Relevant references were identified through a systematic search of English-language publications from 2020 to 2024. The search strategy employed Boolean logic and a combination of keywords, including “Dengue”, “Dengue virus”, “antiviral peptide”, “AVP”, “generative adversarial network”, “variational autoencoder”, and “deep generative learning”. The following search string was applied: (“Dengue” OR “Dengue virus”) AND (“antiviral peptide” OR “AVP”) AND (“generative adversarial network” OR “variational autoencoder” OR “deep generative learning”). Original research articles were retrieved primarily from PubMed and Google Scholar and selected based on their relevance to the application of deep generative models (DGMs) in the discovery of DENV-specific AVPs.

The search strategy initially identified 14 potentially relevant records. Subsequently, abstract screening was conducted based on predefined inclusion and exclusion criteria. Studies were included if they employed DGMs to generate peptide candidates for AVPs or anti-DENV activity. Exclusion criteria involved studies applying deep generative learning to unrelated tasks, such as peptide secondary structure generation and discrimination, peptide classification, or peptide-based biomarker prediction for clinical diagnostics. Following this screening process, 10 original research articles met the inclusion criteria and were retained for analysis, while 4 articles were excluded due to their lack of relevance to the survey objectives.

In addition, 19 records identified from previous versions of the review were included, bringing the total to 29 articles that were subjected to detailed analysis, with key information extracted for further evaluation. This review was conducted following the PRISMA 2020 guidelines [12]. A PRISMA flow diagram is included in Figure 1 to illustrate the study selection process. As this review focuses on deep learning models applied to anti-DENV research using in silico data sets and computational architectures, it does not meet the eligibility criteria for registration in PROSPERO, which is limited to health-related outcomes. Furthermore, due to the limited number and heterogeneity of curated references, a literature review approach was deemed more appropriate than a systematic review, particularly for emerging fields characterized by scarce or fragmented data. Nonetheless, in adherence to the journal’s submission protocol, the word “systematic” in the title must remain unchanged. Consequently, a review protocol has not been registered in the PROSPERO database.

Additionally, the PubMed and Google Scholar databases were also used to identify publications on the virological characteristics of the DENV, the mechanisms of action of AVPs, and the architectures of deep generative learning models. Additional insights have been thoroughly documented to enhance the comprehensiveness of the report.

## 3. Background

### 3.1. Overview of DENV

The DENV is a member of the *Flaviviridae* family, which encompasses over 70 major human pathogens, predominantly affecting inter-tropical regions that are home to approximately 3.9 billion people. DENV is an arbovirus primarily transmitted to humans through mosquito bites, particularly by species of the *Aedes* genus. The main vectors are *Aedes (Stegomyia) aegypti*, with occasional transmission by *Aedes (Stegomyia) albopictus*. The virus has four distinct serotypes: DENV type 1 (DENV-1), DENV type 2 (DENV-2), DENV type 3 (DENV-3), and DENV type 4 (DENV-4) [3]. The four serotypes of the DENV (DENV1–4) share a nucleotide sequence homology of 65–70%, indicating a close genetic relationship [13]. A fifth serotype of the DENV (DENV-5) was first identified in 2007 in the blood of a patient from Sarawak, Malaysia [14]. The emergence of the new DENV serotype (DENV-5) may result from genetic recombination, natural selection, and genetic bottlenecks. Unlike the other four DENV serotypes, DENV-5 triggers a unique antibody response [15]. DENV-5 has been acknowledged in the scientific community as a potential fifth DENV serotype but was not included by the International Committee on Taxonomy of Viruses (ICTV) among classified species and thus is not formally recognized as a DENV type [16]. All DENV serotypes are capable of causing infections in humans.

In addition, DENV is an enveloped virus classified under the *Flaviviridae* family [17]. It possesses a single-stranded, positive-sense RNA genome with a length of approximately 11 kilobases (kb) [3]. The single predictable polyprotein is translated by the genome that is used by the virus-encoded protein into seven non-structural proteins and three structural proteins, which are defined as follows: nonstructural gene 1 (NS1), nonstructural gene 2A (NS2A) and nonstructural gene 2B (NS2B), nonstructural gene 3 (NS3), nonstructural gene 4A (NS4A), nonstructural gene 4B (NS4B), nonstructural gene 5 (NS5), and the structural proteins, including capsid protein (C), envelope protein (E), and membrane protein (M) [18]. Figure 2 depicts the structure of the enveloped DENV.

The pathogenesis of DENV is highly complex, involving multiple interrelated factors, including both viral and host determinants. Key contributors include the NS1 viral antigen, DENV genome variation, antibody-dependent enhancement (ADE), and host genetic factors [3,13]. The severe clinical manifestations of dengue in humans are largely attributed to the synergistic interactions of these elements. These factors pose significant challenges to the effective treatment and management of DENV infection. A summary of these contributing factors is provided in Figure 3.

Although NS1 is widely recognized for its role in DENV pathogenesis, it is not the only viral protein contributing to disease morbidity and virulence. Structural proteins such as the membrane precursor (prM/M) and the capsid (C) have also been implicated in modulating disease severity and mortality through their intrinsic disorder characteristics. Goh et al. (2019) [19] reported a significant correlation between the predicted intrinsic disorder (PID) percentage of the C protein and fetal morbidity, as demonstrated by a one-way ANOVA (R^2^ = 0.80, F = 16, *p* < 0.01). Notably, the strongest associations emerged from regression analyses where prM (r^2^ = 0.78, *p* < 0.01) or M (r^2^ = 0.91, *p* < 0.01) were used as independent variables, with the C protein and case fatality rates (CFRs) as co-explanatory and dependent variables, respectively. These findings suggest that higher levels of intrinsic disorder in the C and M proteins are positively correlated with increased virulence in flaviviruses [20].

### 3.2. Epidemiology of DENV

The distribution of the serotypes varies in different countries, but serotype-2 has been reported as the predominant serotype in recent years in some countries, including Pakistan, Sri Lanka, Indonesia, India, Thailand, Germany, and France, followed by serotypes 1 and 3 [3]. DENV-2 was the predominant strain during 1973–1986, DENV-1 and DENV-2 during 2000–2010, and DENV-3 during 2000–2002 and 2008–2010. The most common genotype was DENV-1, including the strains isolated during 2015–2017. DENV-2 and DENV-4 were the main circulating serotypes during 2015–2017 [21]. A cross-sectional study was conducted on 96 DENV-infected children admitted to Can Tho Children’s Hospital in Vietnam between October 2022 and March 2023. Among the detected serotypes, DENV-2 was the most prevalent, accounting for 71.87%, followed by DENV-1 (23.96%) and DENV-4 (4.17%). No cases of DENV-3 were identified. DENV-2 was associated with a higher incidence of mucous membrane hemorrhages and gastrointestinal bleeding compared to the other serotypes [22].

### 3.3. Dengue Treatment and Vaccination

Currently, no specific antiviral therapy is available for DENV infection. In most cases, dengue fever resolves spontaneously without the need for targeted treatment. Supportive care, such as fluid replacement, analgesics, and bed rest, is the standard approach. Although no antiviral drug is known to cure dengue, fever can be effectively managed with acetaminophen. It is essential to carefully monitor and manage severe cases of dengue to prevent complications. In May 2024, WHO prequalified a new dengue vaccine (TAK-003), adding to the existing CYD-TDV vaccine developed by Sanofi Pasteur [23]. This highlights the potential of complementary therapeutic approaches, particularly AVPs, as a promising avenue for further investigation.

### 3.4. Drug Development Efforts Targeting DENV and Related Flaviviruses

The challenge in controlling DENV infection arises from several factors, including the existence of four distinct viral serotypes, each capable of causing severe disease independently, limited understanding of its pathophysiology, the absence of a specific treatment, and difficulties in managing vector populations [24]. Numerous antiviral drug candidates have failed to progress to clinical trials due to unfavorable physicochemical and pharmacokinetic properties. The use of non-human primates in antiviral testing is rare due to high costs, breeding challenges, and their inability to fully mimic severe human dengue symptoms [25].

In addition to DENV, other mosquito-borne flaviviruses such as West Nile virus (WNV) and Zika virus (ZIKV) have also caused major outbreaks worldwide in recent years, highlighting the pressing need for broad-spectrum antiviral strategies [26]. DENV, WNV, and ZIKV are closely related members of the *Flaviviridae* family. Although their genetic material is RNA, they exhibit distinct structural and functional characteristics that influence their interactions with the host immune system. For example, ZIKV shares approximately 57% amino acid sequence identity with WNV and 54–58% with DENV serotypes 1–4 [27]. Despite these similarities, significant differences are observed in key viral proteins, particularly the envelope (E) protein and non-structural proteins such as NS1, NS4B, and NS5, which contribute to variations in viral infectivity, immune evasion, and pathogenic potential [26]. Notably, the E protein, a principal component of the viral envelope, displays up to 40% amino acid sequence homology across flaviviruses [28].

In light of the continuing global threat posed by viral pandemics, there is an urgent demand for the development of novel therapeutic agents. Efforts have been reported toward the design of modified peptidic and non-peptidic compounds, natural products, and fragment-based inhibitors (typically with molecular weights under 300 Da) targeting both structural and non-structural viral proteins [29]. Current strategies are largely centered on repurposing structure-based drug design approaches, particularly those aimed at inhibiting key viral enzymes such as the NS3 protease [30,31] and NS5 RNA-dependent RNA polymerase [32,33]. Despite these efforts, the identification of effective antiviral agents against DENV and other flaviviruses has remained largely unsuccessful. These factors underscore the urgent need for innovative strategies, including the integration of artificial intelligence, to overcome the limitations of conventional antiviral discovery pipelines.

### 3.5. Potential of AVP Against the DENV

Therapeutic peptides are a unique class of pharmaceuticals, composed of amino acid sequences with molecular weights ranging from 500 to 5000 Da. They are widely applied in various therapeutic areas, including urology, respiratory diseases, pain management, oncology, metabolic disorders, cardiovascular conditions, and antimicrobial therapy [34].

As natural amino acid-based therapeutics, therapeutic peptides have two intrinsic drawbacks: membrane impermeability and poor in vivo stability, which represent major stumbling blocks for peptide drug development. Natural AVPs have several advantages over traditional chemical compounds. These include high specificity and effectiveness, low toxicity (since the final breakdown products are amino acids), and biodegradability by peptidases (which prevents accumulation in organisms). Additionally, they can inhibit large surface area interactions (such as binding proteins), have low molecular weight, and, most importantly, can exert broad-spectrum activity against different viruses. However, there are some limitations to using these compounds as antivirals. These include a short half-life (rapid blood clearance), potential immunogenicity, complex modes of action, high production costs, and membrane impermeability (which depends on multiple factors, such as length and amino acid composition). Other drawbacks include solubility issues and low oral absorption [34,35]. The advantages and limitations of AVPs compared to chemical molecules are analyzed and illustrated in Figure 4.

### 3.6. Antiviral Discovery Supported by DGMs

There are multiple pathways for peptide development, including peptides that mimic human hormones, rational design of peptides based on protein–protein interactions, and discovery of peptide drug candidates through phage display [34] or AI-based therapeutic peptide development [36,37].

Deep learning techniques, an AI-driven approach, are divided broadly into three major categories (Figure 5): (i) deep networks for supervised or discriminative learning; (ii) deep networks for unsupervised or generative learning; and (iii) deep networks for hybrid learning combining both these and relevant other techniques [38]. Additionally, Figure 6 illustrates the pivotal role of deep learning in peptide analysis, highlighting its application across three core domains: classification, property prediction, and generation [39]. These computational approaches enable the precise identification of functional peptides, predictive modeling of their biochemical properties, and the rational design of novel sequences, collectively advancing peptide-based therapeutic discovery. Among these, common deep neural network techniques for DGM (or unsupervised learning) include GANs, autoencoders (AEs), restricted Boltzmann machines (RBMs), and deep belief networks (DBNs), along with their variants, which are summarized in Table 1 [38,40,41].

DGMs have become a key area in deep learning, recognized for their ability to learn meaningful representations with minimal labeled data. Their efficiency in training on limited unlabeled samples enhances adaptability across various real-world applications. In recent years, two new DGMs have evolved, namely VAEs and GANs [42]. Building upon the foundations discussed above, the application of DGMs in the development of AVP candidates, particularly those specifically targeting the DENV (anti-DENV peptides or ADPs), is a highly promising and emerging field of research.

## 4. Findings

### 4.1. Existing AVP Databases

Although publicly available protein informatics datasets with labeled protein activity exist, their size remains limited, particularly when contrasted with the vast datasets accessible in other scientific domains [43]. Peptide databases serve as invaluable repositories for studying the bioactivity of peptides, particularly in antimicrobial and antiviral research. A comparative assessment of major peptide databases reveals substantial variation in the proportion of AVPs across different datasets. Table 2 summarizes the AVP datasets recorded in our survey. It is evident that AVPs constitute a smaller proportion of AMP datasets. Notably, one dataset, ADPDB, contains 606 AVPs specifically targeting the DENV, providing a valuable data foundation for future research.

### 4.2. Generative Deep Learning in AVP Discovery

Based on a comprehensive analysis of the literature retrieved from the PubMed database, we have identified a range of DGMs employed in the discovery of antimicrobial and AVPs. The AI tools and corresponding DGMs employed in these studies are summarized in Table 3. Notably, no studies utilizing DGMs for the design of AVPs against the DENV have been identified. This highlights a significant research gap and underscores the potential of this approach in our future study.

### 4.3. Peptide Data Representation for DGMs

Feature representation plays a pivotal role in the input processing of generative deep models. The selection of an appropriate encoding approach is critical, as it determines the model’s ability to generalize and extract meaningful patterns from peptide data. However, not all representations mentioned are suitable for generative tasks. An additional constraint that applies to sequence-based feature representations used in peptide generation tasks is the necessity of backward mapping to reconstruct the newly generated sequence. For instance, global properties such as amino acid composition have been widely used to represent peptides in property prediction tasks. However, in generative tasks, an inverse mapping that can transform a peptide represented by global properties back into a sequence does not currently exist [43]. As a result, these representations are often inadequate for generative models due to their incomplete sequence information.

Commonly employed peptide encoding methods in generative tasks include one-hot encoding, learned embeddings, or even direct sequence representation [43,64,65]. Additionally, structure-based representations, such as molecular graph representations and contact maps, along with 3D coordinate-based representations, are widely utilized in generative tasks [65]. Table 4 provides examples of common generative models, including VAEs and GANs, highlighting data representation methods in peptide design. These insights enhance understanding and inform the selection of promising representation techniques for our future research.

### 4.4. Evaluating Peptide Generation Models

Wet-lab experiments are the gold standard for testing peptide performance due to their accuracy, but they are resource-intensive and time-consuming. Therefore, computational models have advanced such that they can evaluate generated peptides effectively. Prediction tools with high-performance metrics assess peptide function likelihood or provide experimentally validated data. Molecular dynamics simulations elucidate peptide–biomolecule interactions, offering insights into stability and binding affinity [65]. Other quantitative evaluation methods are also of interest, including the following: (1) Statistical analysis of sequences: To evaluate generative model performance, Pearson correlation, cosine similarity, and Euclidean distance quantify alignment between generated and original sequences. BLEU and perplexity (PPL) scores assess reconstruction fidelity. Novelty is measured by the proportion of unique peptides absent from the training set, indicating potential overfitting. Diversity among generated peptides is essential for assessing drug candidate potential and detecting mode collapse. (2) Various tools are employed to estimate physicochemical properties, including sequence length, molecular weight, amino acid frequency, net charge, aromaticity, hydrophobicity, and hydrophobic moment from generated sequences. These properties are then statistically analyzed and compared with the training data to evaluate the model’s learning performance. (3) Specific functions are closely linked to structural features. Secondary structure analysis (e.g., α-helix, β-sheet, and random coil) can be assessed using methods like GOR IV to evaluate helicity in designed AVPs.

### 4.5. Potential of AVPs Against DENV

Researchers have explored multiple strategies to inhibit the DENV, primarily by targeting its structural and non-structural (NS) proteins. Peptides engineered to target these key viral components exhibit diverse mechanisms of action, both in in vitro and in silico assays. Specifically, peptide candidates targeting structural proteins—capsid (C), pre-membrane (prM/M), and envelope (E)—have the potential to interfere with virus–host interactions, thereby preventing viral attachment, membrane fusion, and entry into host cells. Meanwhile, NS proteins, which are integral components of the viral replication machinery, present critical targets for antiviral intervention. Peptides designed to disrupt NS protein function can effectively impair viral genome replication and assembly, offering a promising approach to combat DENV infection [6]. Lists of AVPs designed against both structural and non-structural DENV proteins are summarized in Table 5 and Table 6.

The findings indicate that DENV serotype-2 has garnered the most research attention. Among structural proteins, the envelope (E) protein has been the primary focus for peptide-based inhibition studies. In contrast, NS2B-NS3 has emerged as a key target for discovering inhibitors against non-structural (NS) proteins, with fewer studies investigating NS5. Notably, all in silico experiments have exclusively utilized molecular docking, underscoring the potential of DGM for advancing AVP design and discovery in DENV research.

## 5. Discussion

### 5.1. Generative Adversarial Networks

GANs are a type of deep learning model used to generate new data that closely resembles real data. GANs operate based on an adversarial game or a minimax two-player game between the Generator (G) and Discriminator (D). The generator’s role is to create synthetic data that look as realistic as possible. The discriminator’s role is to distinguish between real/original data (from an actual dataset) and fake/generated data (generated by the generator) [65]. The architecture of GANs is illustrated in Figure 7. The discriminator assesses whether a given sample is real or fake by assigning a probability score. A high probability means the sample is likely real, while a low probability indicates it is fake. The objective is to train both the generator and discriminator until the discriminator can no longer reliably distinguish between real and generated data, assigning a probability of 0.5 to both. At this point, the generator has successfully learned the true data distribution, reaching an optimal state [42]. GANs have achieved remarkable success; however, they still encounter significant challenges, particularly mode collapse [42,97], which impedes stable training and limits the generation of diverse, high-quality samples. Mode collapse is a critical factor contributing to the instability of GAN training. This phenomenon arises when the generated outputs are constrained to a limited set of similar or identical samples, thereby failing to capture the full diversity of the training data distribution [98]. As a result, various GAN variants have been introduced to tackle mode collapse, such as Wasserstein GAN with gradient penalty (WGAN-GP) [65,99] and conditional GANs (CGANs) [100].

### 5.2. Variational Autoencoders

AEs are a type of feed-forward neural network designed for unsupervised learning, functioning without recurrence. Their primary objective is to learn a compressed representation of input data and subsequently reconstruct the original input from this reduced form. The dimensionality of the output vectors remains consistent with that of the input vectors [42]. An AE comprises three fundamental components: the encoder, the latent space (code), and the decoder. The encoder maps the input data into a lower dimensional representation known as the latent space. The transformation captures and compresses essential features while discarding redundant information. The latent space (code) stores the compressed version of the input, preserving meaningful and relevant information. The decoder reconstructs the original input from the latent space with minimal reconstruction loss, aiming to retain the essential characteristics of the initial data [42].

Limitations of traditional AEs are as follows: Conventional AEs compress input data into a latent representation, which serves as a lower dimensional encoding of the original input. However, this latent space lacks a well-structured probability distribution, making it difficult to sample new meaningful data points. Consequently, traditional AEs struggle to generate novel data, as they do not provide a structured way to explore the latent space.

VAEs are an improved version of traditional AEs that incorporate probabilistic modeling using variational Bayesian inference. Unlike standard autoencoders, which encode input data into fixed points in a latent space, VAEs represent the latent space as a probability distribution (typically Gaussian). Instead of directly mapping inputs to a single encoding, VAEs learn to estimate the meaning and variance of a distribution, allowing for controlled sampling of latent vectors. In a VAE, the encoder is also referred to as the recognition model, while the decoder is known as the generating model. These two models are interconnected but parameterized independently [42]. The architecture of VAEs is given in Figure 8.

Both VAEs and AEs generally consist of an encoder and a decoder. However, an AE focuses on representation learning by extracting meaningful features from raw data, whereas VAEs aim to generate new data by sampling from a latent variable *z* and reconstructing *x* [39,65]. The loss function of VAEs comprises two key components: the reconstruction loss and the Kullback–Leibler (KL) divergence. By optimizing this loss, the VAEs capture the underlying latent representations, enabling the decoder to generate new data. These newly generated samples resemble the original data but are not identical to it [39]. The encoder in VAEs can be implemented using different architectures, such as Recurrent Neural Networks (RNNs) or Convolutional Neural Networks to learn latent representations, while the decoder is typically an RNN when used for sequential data generation [65].

### 5.3. GAN Architectures in AVP Discovery

Based on Table 3, it is evident that GAN-based frameworks are commonly employed for the generation of AVPs. These include bidirectional conditional GANs (BiCGANs), Feedback GANs, and LeakGANs, each offering distinct advantages in capturing sequence patterns and optimizing peptide features. To the best of our knowledge, however, no studies to date have reported the application of GAN frameworks specifically for the discovery or development of ADPs.

AMPGAN v2 is a bidirectional conditional GAN (BiCGAN) designed for the generation of AMPs, including AVPs. It leverages a generator–discriminator framework to learn data-driven priors for peptide generation, guided by conditioning variables that enable controlled synthesis of antimicrobial sequences. A key innovation in this architecture is the incorporation of a bidirectional component through a learned encoder, which maps real peptide sequences into the generator’s latent space. This bidirectional mapping facilitates iterative refinement and targeted manipulation of candidate peptides, supporting more efficient optimization of desired biological properties [61]. From this generative model, a total of 5000 AMP candidates were produced. To evaluate the reliability and biological relevance of the framework, the authors conducted a comprehensive comparative analysis of physicochemical properties between real and generated peptide datasets. The results revealed a high degree of similarity in amino acid composition, with deviations of less than 1% for most of the 20 standard amino acids. Furthermore, sequence diversity analysis using the Gotoh global alignment algorithm demonstrated that the generated AMPs were largely novel compared to those in the training set. To further assess functional potential, the sequences were subjected to in silico validation via the CAMPR3 platform [101], employing four distinct machine learning classifiers: Support Vector Machine (SVM), Random Forest (RF), Artificial Neural Network (ANN), and Discriminant Analysis (DA). The classifiers predicted that 79.85%, 88.36%, 88.24%, and 83.71% of the generated sequences, respectively, were likely to exhibit antimicrobial activity.

Another system that was applied for AVP generation was Feedback-AVPGAN, which combined the WGAN-GP and a feedback module to address the challenge of limited data on known AVPs. This feedback mechanism allowed the model to iteratively learn not only from existing AVPs but also from synthetic peptides generated during training. Although these generated peptides were computational predictions, they contributed to expanding the pool of potential AVPs. Using this approach, 205 peptide sequences were generated. To assess their similarity to known AVPs, pairwise alignment methods were used. Furthermore, structural similarity was evaluated using predicted Local Distance Difference Test (pLDDT) scores, and solubility predictions were also performed for the candidate sequences [63].

LeakGAN is a GAN architecture designed to address the challenge of sparse feedback signals typically encountered during sequence generation. Unlike standard GANs, LeakGAN incorporates concepts from hierarchical reinforcement learning to improve information flow from the discriminator to the generator. Specifically, it features a hierarchical generator composed of two modules: a high-level *Manager* and a low-level *Worker*, both implemented using long short-term memory (LSTM) networks. This hierarchical design enables more informative and fine-grained guidance during training, allowing the generator to produce more coherent and biologically relevant sequences [102]. The PandoraGAN tool, which utilizes the LeakGAN architecture, was employed to generate AVP candidates using a validation strategy based on key physicochemical properties. Specifically, peptides were filtered based on criteria including a net charge at pH 7.4 ≥ −1, GRAVY score > −1, sequence length between 7 and 30 amino acids, and secondary structure features such as a helix probability > 0.35 and a strand probability < 0.3. Next, the generated sequences underwent further cross-validation using established in silico antiviral classification tools, such as AVPpred (http://crdd.osdd.net/servers/avppred/ (accessed on 18 February 2025) [103] and Meta-iAVP [104]. Remarkably, over 70% of the PandoraGAN-generated sequences were predicted to be AVPs with a confidence score greater than 0.9. Notably, the study also focused on the ability of PandoraGAN to learn sequence-specific features from a particular viral family, Flaviviridae. The training data set for this purpose included AVPs with molecular weights below 3000 Da, shorter sequences (<20 amino acids), and a high helical propensity (>0.3). Analysis showed that the generated peptides contained motif patterns similar to those found in the input data set, with conserved sequences such as XWLRD appearing in both [62]. These preliminary findings highlight the promising potential of GAN-based techniques in discovering novel AVPs against DENV, a member of the *Flaviviridae* family.

### 5.4. VAE Architectures in AVP Discovery

To the best of our knowledge, no studies have yet reported the use of VAEs for the specific generation of AVPs or ADPs. While VAEs have demonstrated promising capabilities in various AMP design tasks [54,105], their application in the context of AVP and ADP discovery remains largely unexplored. This gap highlights an important opportunity for future research to investigate the potential of VAE-based frameworks in designing novel peptide therapeutics against viral targets, including DENV.

### 5.5. Opportunities and Limitations of DGMs in DENV-Specific AVP Discovery

In general, traditional peptide discovery relies on iterative screening, synthesis, and optimization, which are time-consuming, costly, and limited in chemical space exploration. AI-driven methods, specifically DGMs, enhance efficiency by utilizing machine learning and virtual screening to predict peptide activity, prioritize candidates, and streamline validation. This approach accelerates discovery, reduces costs, and broadens the scope of potential peptides. Key insights are summarized in Table 7 [36].

Recently, the release of a dedicated database for ADPs (ADPDB, 2024) [52] provides a timely and promising foundation for the application of DGMs in the discovery of AVPs targeting DENV. In addition, synthetic data generation and augmentation strategies offer potential solutions to the longstanding issue of data scarcity in this domain. Looking ahead, integrating DGM-based design pipelines with high-throughput in vitro screening platforms represents a key opportunity to accelerate the peptide discovery-to-validation process. Bridging the gap between in silico generation and wet-lab validation will be essential for translating DGM-derived candidates into viable therapeutic agents.

Despite their potential, the application of GANs and VAEs in the discovery of ADPs faces several critical limitations. One of the primary challenges lies in the need for large and high-quality datasets to enable these models to learn meaningful sequence patterns. However, DENV-specific AVPs remain severely underrepresented in public repositories, which restricts model training and limits generalizability. Although the updated ADPDB includes 606 anti-DENV peptides, this figure accounts for only ~22% of the 2683 peptides in the broader AVPdb. Moreover, the ADPDB consists solely of positive samples, including 593 experimentally validated peptides and 13 predicted through computational methods. The AVPdb, in comparison, includes just 38 experimentally confirmed anti-DENV peptides. In total, the number of experimentally verified positive sequences available for model training is limited to only 644, which remains relatively modest. A further major obstacle is the lack of experimentally validated non-active (negative) peptide sequences, not only for DENV but for other viral pathogens as well, which hampers the development of robust discriminative models [48].

In addition, most peptides generated by GANs and VAEs have been evaluated exclusively through in silico analyses, with little or no validation via in vitro or in vivo experiments. For instance, PepGAN synthesized six AMP candidates, but only one demonstrated a minimum inhibitory concentration (MIC) value of 3.1 μg/mL, outperforming the widely used antibiotic ampicillin (MIC 6.25 μg/mL) [58]. As previously mentioned, we have not identified any DGM-based studies specifically targeting anti-DENV peptide design, and therefore the potential of these models for peptide synthesis in this context remains unconfirmed.

Furthermore, GAN-based models are prone to mode collapse, frequently resulting in low-diversity peptide outputs. This phenomenon significantly undermines the discovery of structurally and functionally novel candidates, which is critical for the development of effective antiviral therapeutics.

### 5.6. Proposed DGM-Based Workflow for AVP Design Against the DENV

Leveraging key insights from this report and the relevant literature, we have developed a systematic and robust workflow for the discovery of AVPs targeting the DENV using DGMs (Figure 9). This framework establishes a strategic and data-driven approach to enhance the exploration, characterization, and optimization of AVPs, accelerating the discovery process and deepening our understanding of peptide-based therapeutics for DENV intervention.

### 5.7. Future Directions

GANs have shown considerable promise in the design of AVPs, positioning them as strong candidates for the discovery and development of ADPs. Advanced GAN variants that address limitations such as mode collapse, especially WGAN-GP and CGANs, may offer robust frameworks for future investigations. Additionally, VAEs, which have been effectively utilized in related peptide and protein design tasks, hold potential for adaptation in the identification of peptide candidates targeting the DENV.

To enhance real-world applicability, future models should prioritize virus-specific designs, particularly those targeting key DENV proteins (i.e., NS1 or E proteins). In parallel, deep neural network-based predictors can be developed to assess the antiviral potential of generated peptides, enabling a hybrid framework that integrates DGMs with in silico prediction pipelines. Such integration would facilitate the identification and prioritization of high-potential ADP candidates for downstream validation.

In addition to binary classification of anti-DENV activity, regression-based approaches could be leveraged to predict quantitative indicators such as IC_50_ values, thereby enhancing the prioritization and efficiency of downstream experimental validation. Looking ahead, these generative frameworks may not only accelerate DENV-specific peptide discovery but also be extended to a broader spectrum of viral pathogens, supporting the development of versatile, next-generation antiviral therapeutics.

To provide more concrete future directions, subsequent studies could utilize generative models such as GANs or VAEs to design novel peptide candidates based on validated and high-quality data sets. These candidates should undergo initial screening through robust in silico prediction tools, followed by in vitro assays, where applicable, to assess their potential antiviral or anti-DENV activity. Peptides demonstrating favorable properties should be further evaluated for drug-likeness and synthetic accessibility to determine their suitability for practical application. Selected candidates may then be synthesized and subjected to preclinical testing to evaluate their efficacy and safety. Furthermore, future research could investigate how AI-assisted peptide discovery might be integrated into public health initiatives, particularly in the context of emerging infectious diseases and the rapid development of antiviral therapeutics.

## 6. Conclusions

This article underscored the substantial disease burden associated with viral infections, with a particular emphasis on DENV infection. The lack of specific antiviral treatments for dengue has highlighted the critical need for alternative therapeutic strategies, among which AVPs represent a promising avenue. Our study provides a comprehensive overview of the application of DGMs, including GANs and VAEs, in AVP discovery, as well as a critical assessment of available data sets and the limitations of current generative approaches.

Notably, in silico and in vitro evidence support the potential of AVPs as viable therapeutic candidates. However, major challenges remain, particularly the scarcity of high-quality annotated data sets and the limited experimental validation of computational outputs. Overcoming these barriers will require the development of integrated discovery pipelines that connect DGM-based peptide generation with downstream experimental screening. Incorporating virus-specific biological insights into model training may further enhance prediction accuracy and clinical relevance.

These insights lay the groundwork for advancing AI-assisted peptide design, not only in the context of DENV but also for broader applications against emerging viral pathogens. Future work should prioritize bridging the gap between computational design and experimental validation, thereby enhancing the translational potential of AVP and ADP development.

## Figures and Tables

**Figure 1 ijms-26-06159-f001:**
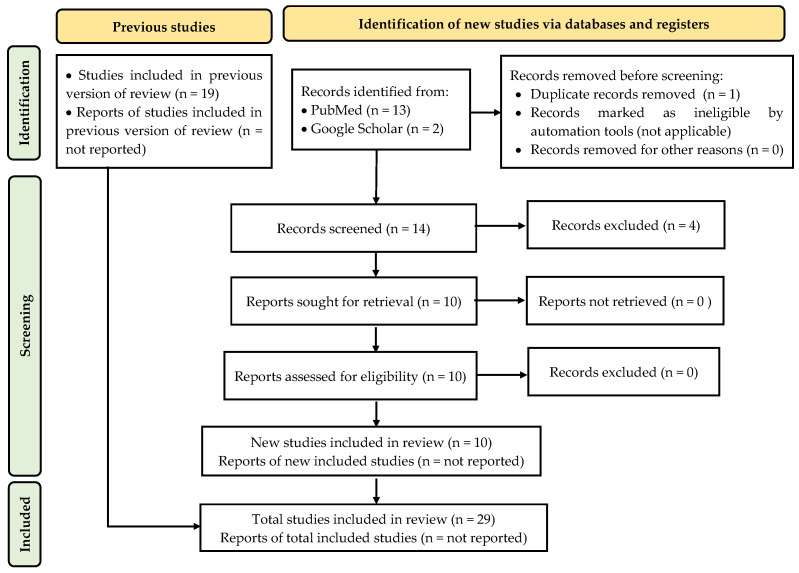
The PRISMA flow diagram of the search strategy used in this study.

**Figure 2 ijms-26-06159-f002:**
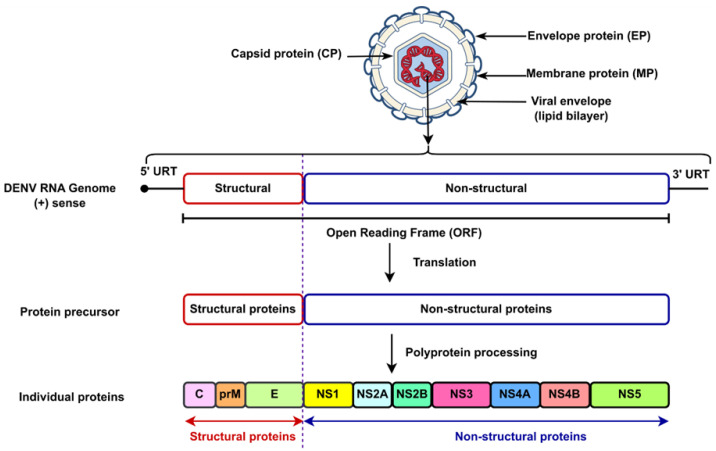
The structure of the enveloped DENV.

**Figure 3 ijms-26-06159-f003:**
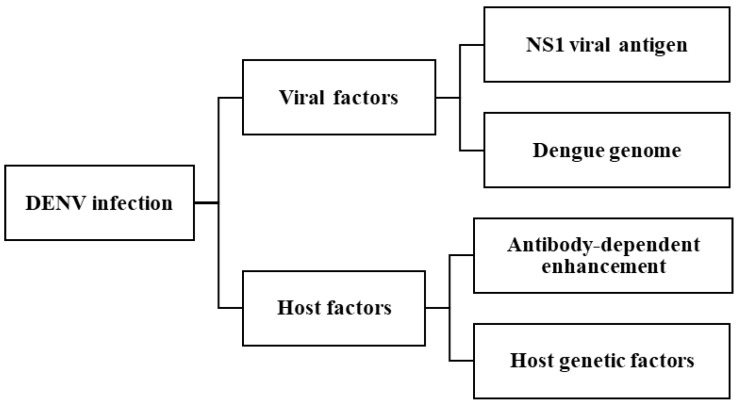
Key factors influencing DENV infection.

**Figure 4 ijms-26-06159-f004:**
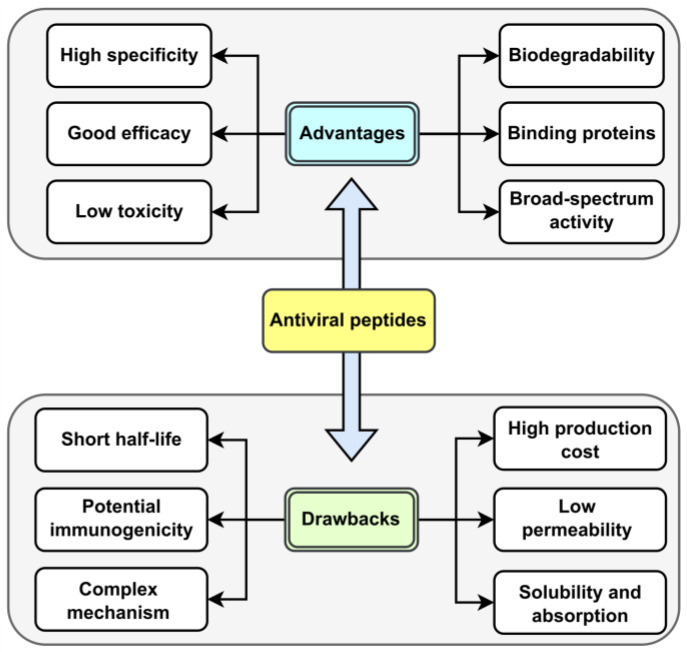
Advantages and drawbacks of AVPs and chemical molecules.

**Figure 5 ijms-26-06159-f005:**
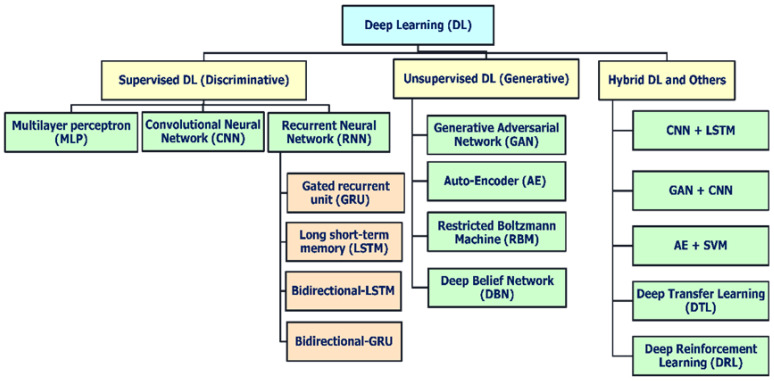
Overview of deep learning models and architectures.

**Figure 6 ijms-26-06159-f006:**
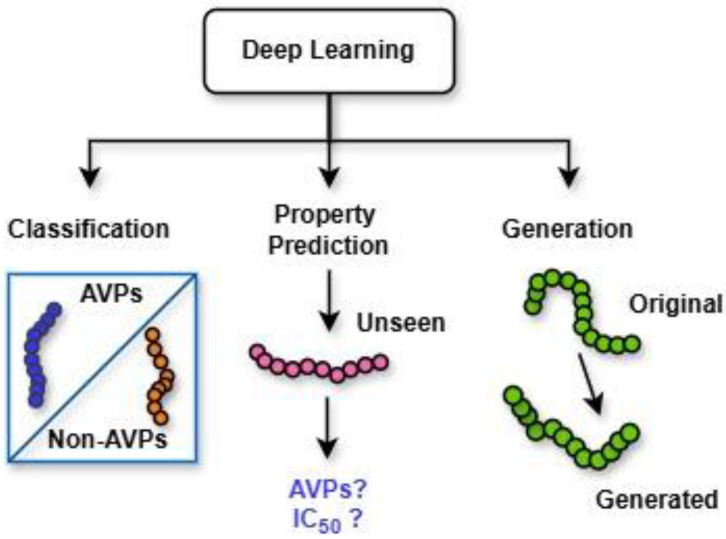
Applications of deep learning in peptide analysis.

**Figure 7 ijms-26-06159-f007:**
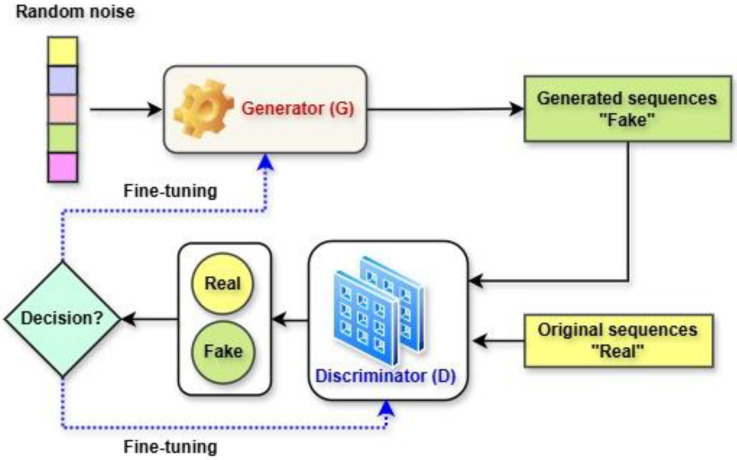
GAN architecture.

**Figure 8 ijms-26-06159-f008:**
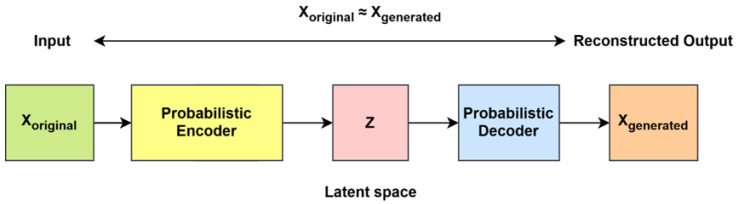
VAE architecture.

**Figure 9 ijms-26-06159-f009:**
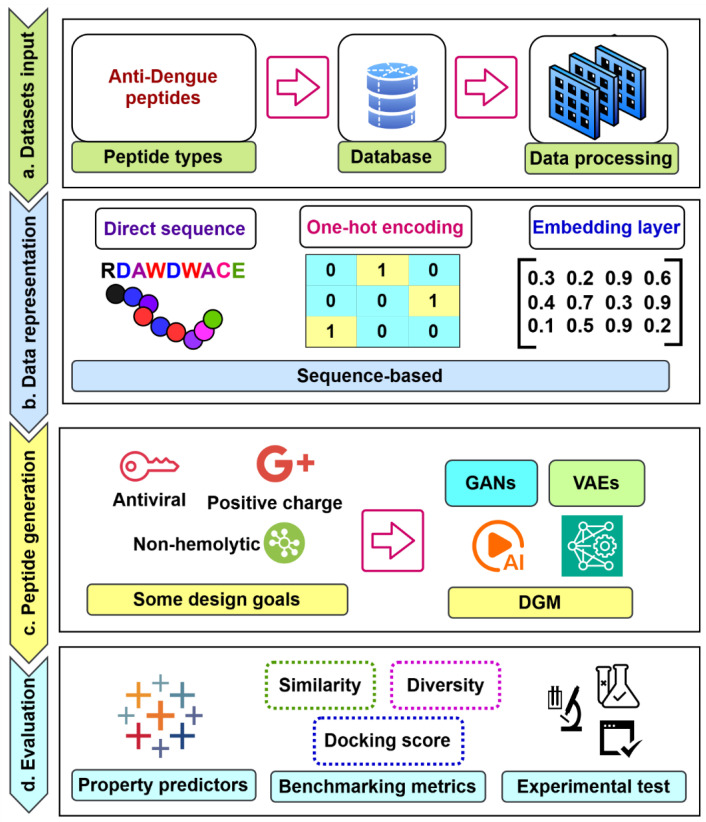
Workflow for the generation of ADPs using DGMs.

**Table 1 ijms-26-06159-t001:** DGMs and their variants.

DGMs	Their Variants
GANs	Traditional GANs Wasserstein GANs Signal-augmented self-taught learning
AEs	Stacked AEs Variational AEs (VAEs) Convolutional AEs
RBMs	Shallow RBMs Convolutional RBMs
DBNs	Simple DBNs Conditional DBNs

**Table 2 ijms-26-06159-t002:** A summary of peptide datasets containing AVPs.

Database	Labels	Number of Peptides *	AVPs (%)
ADP [44]	Antimicrobial and various endpoints	5099	4.94
YADAMP [45]	Antimicrobial (antibacterial)	2133	0
LAMP [46]	Antimicrobial and antitumor activity	5547	20.9
AVPdb [47]	Antiviral	2683	100
DBAASP [48]	Antimicrobial and various endpoints	61,894	2.45
CAMPR4 [49]	Antimicrobial	21,441	5.75
DRAVP [50]	Antiviral	1986	93.5
DRAMP 4.0 [51]	Antimicrobial and various endpoints	30,260	12.0
ADPDB [52]	Antiviral (DENV)	606	100
dbAMP 3.0 [53]	Antimicrobial and various endpoints	35,518	3.15

* The databases were accessed on 17 February 2025.

**Table 3 ijms-26-06159-t003:** AI tools and DGMs for antimicrobial and AVP design.

Endpoints	AI Tools	DGMs
Antimicrobial	PepVAE	VAEs [54]
	HydrAMP https://hydramp.mimuw.edu.pl/ (accessed on 18 February 2025)	VAEs [55]
	-	VAEs [56]
	-	Wasserstein GANs with gradient penalty (WGAN-GP) [57]
	PepGAN	GANs [58]
	FBGAN	Feedback GANs [59]
	dsAMPGAN	GANs [60]
Antiviral (multiple viral targets)	AMPGAN v2	Bidirectional conditional GANs [61]
	AI4AVP https://axp.iis.sinica.edu.tw/AI4AVP/ (accessed on 18 February 2025)	Data augmentation by GANs [10]
	PandoraGAN	LeakGANs [62]
	Feedback-AVPGAN	Feedback GANs [63]
Antiviral (targeting the DENV)	No literature found	

**Table 4 ijms-26-06159-t004:** Popular sequence representations used in VAEs and GANs for generative tasks.

Category	Model	Database	Representation
VAEs	CLaSS [66]	UniProt/satPDB, DBAASP, AMPEP	Embedding layer
VAEs	PepVAE [55]	APD, DADP, DBAASP, DRAMP, YADAMP	One-hot
VAEs	HydrAMP [56]	dbAMP, AMP Scanner8, DRAMP6	One-hot
GANs	GANDALF [67]	THPdb	3D coordinate
GANs	PepGAN [58]	APD, CAMP, LAMP, DBAASP	Direct sequence
GANs	AMPGAN v2 [61]	DBAASP, AVPdb, UniProt, AVPdb, AVPred, CAMP, Dramp, APD3, dbAMP	One-hot
GANs	PandoraGAN [62]	dbAMP	Embedding layer

**Table 5 ijms-26-06159-t005:** List of AVPs against DENV structural proteins.

Assays	DENV Serotypes	Sequences	Target
In vitro	All serotypes	MAILGDTAVDFGSLGGVFTSIGKALHQVFGAIY [68,69] (IC_50_ < 10 μM)	E
In vitro	DENV-2	DTRACDVIALLCHLNT [70] 200 µM (~99.3% inhibition)	E
In vitro	All serotypes	AWDFGSLGGVFTSIGKALHQVFGAIYGAA (solubility tag-RGKGR) [71,72] IC_50_ of 0.125 µM (DENV-2) (Fluorescence polarization assay)	E
In vitro	All serotypes	AWDFGSLGGVFTSIGKALHQVF [72] IC_50_ of 0.25 µM (DENV-2) (Fluorescence polarization assay)	E
In vitro	All serotypes	AILGDTAWDFGSLGGVFTSIGKALHQVGAIYGAA [72] IC_50_ of 0.275 µM (DENV-2) (Fluorescence polarization assay)	E
In vitro	All serotypes	AILGDTAWDFGSLGGVFTSIGKALHQVF [72] IC_50_ of 0.25 µM (DENV-2) (Fluorescence polarization assay)	E
In vitro	All serotypes	AILGDTAWDFGSLGGVFTSIGKA [72] IC_50_ of 2 µM (DENV-2) (Fluorescence polarization assay)	E
In vitro	All serotypes	AWDFGSIGGVFTSVGKLVHQVFGTAYGVL (solubility tag-RGKGR) [71] IC_90_: DENV-1: 1.5 µM; DENV-2: 2 µM DENV-3: >6 µM; DENV-4: >6 µM	E
In vitro	All serotypes	AWDFGSVGVLNLSLGKMVHQIFGSAYTAL (solubility tag-RGKGR) [71] IC_90_: DENV-1: 0.1 µM; DENV-2: 2 µM DENV-3: 4 µM; DENV-4: 1.5 µM	E
In vitro	All serotypes	AWDFGSVGGFLTSLGKAHVQFGSVYTTM (solubility tag-RGKGR) [71] IC_90_: DENV-1: 5 µM; DENV-2: 6 µM DENV-3: >6 µM; DENV-4: 6 µM	E
In vitro	DENV-2	RWMWRWHFHRLRLPLYNPGKNKQNQQWP [73] IC_50_ of 8 ± 1 µM	E
In vitro	DENV-2	FWFTLIKTQAKQPARYRFFC [73] IC_50_ of 7 ± 4 µM	E
In vitro	DENV-2	RQMRAWGDYQHGGMGYSC [73] IC_50_ of 36 ± 6 µM	E
In silico (MD)	DENV-2	FPFDHHDRYHIYHWHKRYQH [74] (IC_50_ values were not reported)	E
In silico (MD)	DENV-2	IWWRPRDWPITFIYFWRRW [74] (IC_50_ values were not reported)	E
In silico (MD)	DENV-2	KEYFRRFHCHNHQREWHWH [74] (IC_50_ values were not reported)	E
In silico (MD)	DENV-2	KEKRREWEWRFWERFLYFE [74] (IC_50_ values were not reported)	E
In silico (MD)	DENV-2	RHWEQFYFRRRERKFWLFFW [74] (IC_50_ values were not reported)	E
In vitro	DENV-2	AGVKDGKLDF [75] IC_50_ of 35 μM	E
In vitro	DENV-2	EF [76] IC_50_ of 96 µM	E
In vitro	DVEN-2	GGARDAGKAEWW [77] IC_50_ of ~77–91 µM	E
In vitro	DENV	SVALVPHVGMGLETRTETWMSSGEGAWKHVQRIETWILRHPG [78] IC_50_ of 24–31 µM	prM
In vitro	DENV-1 and DENV-2	pr protein (the precursor fragment of the prM protein) [79] 30 µM (81–85% inhibitions)	prM
In silico (MD)	DENV	NMLKRARNRV [80] Binding forces decreased with 100 µM peptide	C

MD: Molecular docking.

**Table 6 ijms-26-06159-t006:** List of AVPs against DENV non-structural proteins.

Assays	DENV Serotypes	Sequences	Target
In vitro	DENV-2	Phenylacetyl-K-R-R-H [81] IC_50_ of 6.7 µM	NS2B-NS3
In vitro	DENV-2	Benzoyl-n-K-R-R-H (n = norleucine) [81,82,83] IC_50_ of 9.5 µM	NS2B-NS3
In vitro	DENV-2	4-Aminophenylacetyl-K-R-R-H [81] IC_50_ of 11.2 µM	NS2B-NS3
In vitro	DENV-2	4-Phenylphenylacetyl-K-R-R-H [81] IC_50_ of 12.2 µM	NS2B-NS3
In vitro	DENV-3	RPDFC LEPPY TGPKC ARIIR YFYNA KAGLC QTFVY GGCRA KRNMF KSAED CMRTC GGA [83] *K_i_* of 0.026 µM	NS2B-NS3
In vitro	DENV-2	Ac-FAAGRR-CHO [84] *K_i_* of 16 µM	NS2B-NS3
In vitro	DENV-2	Ac-FAAGRR-aketo-SL-CONH_2_ [84] *K_i_* of 47 µM	NS2B-NS3
In vitro	DENV	Ac-RTSKKR-CONH_2_ [85] *K_i_* of 12 µM	NS2B-NS3
In vitro	DENV-3	PCRARIRYGCA [86] *K_i_* of 2.9 µM	NS2B-NS3
In vitro	DENV-2	Bz-Arg-Lys-L-Phg-NH_2_ by the combination of 4-CF_3_-benzyl ether and thiazole cap [87] *K_i_* of 12 nM; EC_50_ of 20 µM	NS2B-NS3
In vitro	DENV-2	Bz-Arg-Lys-L-Phg-NH_2_ by the combination of 4-CF_3_-benzyl ether and thien-2-yl cap [87] *K_i_* of 19 nM; EC_50_ of 7 µM	NS2B-NS3
In vitro	DENV-2	Bz-Arg-Lys-L-Phg-NH_2_ by the combination of 3-OCH_3_-benzyl ether and bithiophene cap [87] EC_50_ of 3.42 µM	NS2B-NS3
In vitro	DENV-2	Bz-Nle-Lys-Arg-Bip [82,88] *K_i_* of 1.16 × 10^4^ nM	NS2B-NS3
In vitro	DENV-2	CKRKC [89] *K_i_* of 0.707 μM	NS2B-NS3
In silico (MD)	DENV-2	AIKKFS [90] Glide energy −80.4 kcal/mol	NS2B-NS3
In vitro	DENV-2	R-Arg-Lys-Nle-NH_2_ with an arylcyano-acrylamide group as N-terminal cap [91] *K_i_* of 4.9 µM	NS2B-NS3
In vitro	DENV-2	Rhodanine-based peptide hybrid bearing a cyclohexyl moiety at the heterocycle [92] EC_50_ of 16.7 µM	NS2B-NS3
In vitro	DENV-2	Thiazolidinedione-based peptide hybrid [92] *K_i_* of 1.5 µM; IC_50_ of 2.9 µM	NS2B-NS3
In vitro	DENV-2	CGKRKSC [93] *K_i_* of 1.4 µM	NS2B-NS3
In vitro	DENV-2	CAGKRKSG [93] *K_i_* of 2.2 µM	NS2B-NS3
In vitro	DENV-2	RGGRLCYCRRRFCVCVGR [94] IC_50_ of 11.7 μM	NS2B-NS3
In vitro	DENV-2	GICRCICGRGICRCICGRPGVPVPVGPVGIHHHHHH [95] IC_50_ of 21.4 μM	NS2B-NS3
In silico (MD)	DENV-2	CQCASKQDKKKSWYCQAKEI [96] ΔG of −24.73 kcal/mol	NS5
In silico (MD)	DENV-2	ETPDCFWKYCV [96] ΔG of −19.04 kcal/mol	NS5

MD: Molecular docking.

**Table 7 ijms-26-06159-t007:** A comparison of conventional and AI-driven peptide exploration techniques.

Aspect	Conventional	AI-Driven
Time and resources	Years of research and high cost	Faster and cost-efficient
Screening	Slow physical testing	Virtual computational analysis
Chemical space	Limited exploration	Broad and systematic exploration
Synthesis and testing	Expensive and complex	Efficient and innovative synthesis
Biological prediction	Trial-and-error experiments	ML-based activity prediction
Peptide design	Manual and intuition-driven	AI-assisted optimization
Sequence design	Chemical synthesis	Deep learning algorithms

## Data Availability

Not applicable.

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
