# Peer review of "Deep Generative Models for the Discovery of Antiviral Peptides Targeting Dengue Virus: A Systematic Review"

_ijms, 2025, doi:10.3390/ijms26136159_

Round 1
Reviewer 1 Report
Comments and Suggestions for Authors
The manuscript entitled “Deep Generative Models for the Discovery of Antiviral Peptides Targeting Dengue Virus: A Systematic Review” addresses a relevant issue: the absence of specific antiviral treatments for dengue, the therapeutic potential of antiviral peptides (AVPs), and the use of deep generative models (DGMs) in their discovery.
Section 1 is appropriate and sufficiently introduces the reader to the clinical, therapeutic, and computational context of the problem.
The manuscript includes numerous recent sources (2020–2024), including peer-reviewed articles and specialized databases. However, Section 2 (Methodology) would benefit from a more detailed description of the inclusion/exclusion criteria and potential biases, in order to strengthen methodological transparency and systematic rigor.
Both the title and abstract indicate that the main focus of the review is the use of DGMs to design AVPs targeting the dengue virus. Within this framework, virological background information should be presented accurately but concisely, without dominating the manuscript or diverting from its central focus.
Sections 3.1 to 3.11, currently included under the Results section, do not reflect outcomes derived from the systematic search, but rather provide a general overview of virological aspects of DENV (epidemiology, serotypes, viral proteins, pathogenic mechanisms, among others). While this information is valuable as background, it is overly developed and creates an imbalance in the manuscript, which should prioritize AVPs and AI models. It is recommended that these contents be condensed, consolidating repeated ideas (e.g., the role of NS1 or serotype variability) into a more concise narrative (1–2 paragraphs per topic), ideally as part of the Background section. This would allow more space for a comparative analysis of generative models, bioinformatics tools, and experimental validation, which represent the core of the review.
Sections 3.12 to 3.21 are aligned with the objectives of the manuscript and provide appropriate context. Since the topics discussed exceed my area of expertise, I will not comment on their relevance, content, or depth of development. I will focus solely on the writing aspects. Some points for revision include:
Section 3.12 does not provide relevant information and could be removed.
Section 3.13 repeats concepts already introduced in the introduction and may be shortened. I suggest reducing it by eliminating lexical and conceptual redundancies. Additionally, Figure 6 mentioned in this section is unclear—it is not evident whether the terms listed on the left refer to chemical molecules and those on the right to antiviral peptides. The arrows add confusion.
Section 3.14 also has limited specific relevance and could be summarized or integrated into another section.
Sections 3.18 and 3.19, although addressing relevant computational techniques (data augmentation, encoding, K-mers, embeddings), exceed the depth required for a systematic review unless explicitly linked to AVPs applied to DENV.
Moreover, in Section 4—particularly subsections 4.1 and 4.2—general definitions and benefits of AVPs and AI are repeated without advancing toward critical analysis. This limits the contribution for an informed reader. It is suggested to enhance the analytical component by comparatively discussing model limitations, their practical applications in the context of dengue, and unexplored opportunities.
Section 4.5 (Future Perspectives) lacks specificity. It would be advisable to more clearly articulate how the findings reviewed could drive concrete developments, such as in silico or in vitro validations, preclinical testing, or clinical integration strategies for AI-generated AVPs.
The critical discussion could be expanded to include reflections on biases and limitations of current models; comparison between reviewed studies; cross-validation; and reproducibility.
Additionally, the tone remains more descriptive than analytical in parts of sections 4.1 and 4.2, where general definitions or already-discussed advantages of AVPs or AI are reiterated instead of addressing their comparative impact, real-world limitations, or direct applications to DENV.
Section 5 (Conclusion) is clear and logically structured, though somewhat lengthy and includes repetitions from the abstract and discussion, with a focus on summarization. A more forward-looking reflection is recommended: for instance, what concrete challenges must be overcome to apply these models to AVPs against DENV? What future steps are advisable for experimental or clinical validation? What short- or medium-term impact could this technology have?
Comments on style and writing:
In the abstract, Latin terms such as in silico, in vitro, in vivo should appear in italics. Abbreviations not used repeatedly in the abstract (e.g., VAEs, GANs) need not be defined there.
Throughout the manuscript, terminological consistency (e.g., AVPs, DGMs, DENV) and uniform acronym usage should be ensured. Acronyms defined in the abstract or introduction should not be redefined or written out again (e.g., L61, L69, and L360/368/371/374/558/638/657/661). Abbreviations introduced in the main text should remain consistent across all sections, including headings, figures, captions, and tables.
The use of unnecessary capital letters should be revised, particularly in section titles and common nouns.
From L52 onward, following editorial guidelines such as ICMJE, AMA, and common scientific writing standards:
Disease names are written in lowercase (e.g., dengue, malaria), unless at the beginning of a sentence.
Virus names, as taxonomic proper nouns, are capitalized: Dengue virus, Influenza A virus, etc.
Virus acronyms are written in uppercase (e.g., DENV, HIV, HBV) and, once established, should be used consistently throughout the manuscript. For instance, “Dengue virus” or “Dengue virus type 1” should not be repeated after “DENV” and “DENV-1” have been introduced.
Specific corrections:
L124–L127: DENV-5 has been acknowledged in the scientific community as a potential fifth dengue virus serotype but was not included by the ICTV among classified species and thus is not formally recognized as a DENV type. It was identified in 2007 from a patient sample in Malaysia and published in 2013. It differs from the other four serotypes by following a sylvatic cycle (transmission between wild animals and mosquitoes) rather than the human-mosquito-human transmission of DENV-1 to DENV-4. Source: https://ictv.global/report/chapter/flaviviridae/flaviviridae/orthoflavivirus
L153: Avoid redundancy. Suggested revision: “The most common DENV-1 genotype was genotype I, including strains isolated between 2015 and 2017.”
L188: Update information: In May 2024, WHO prequalified a new dengue vaccine (TAK-003), adding to the existing CYD-TDV vaccine developed by Sanofi Pasteur. These vaccines and their use are detailed in reference 25 of the manuscript. Source: https://www.who.int/news/item/15-05-2024-who-prequalifies-new-dengue-vaccine
L278: Standardize title formatting (capitalize or lowercase consistently across all section headings).
L461/471: Italicize K-mer and standardize capitalization in “(binary encoding)”.
The remaining corrections are highlighted in the revised manuscript file.
Author Response
Dear Reviewers, we genuinely appreciate all the comments and suggestions, which allowed us to improve the quality of the manuscript significantly. Our responses to the reviewer’s comments are provided below in a point-by-point fashion. All changes to the text in the revised manuscript are made and marked up using the red color highlight. We hope that our revisions and responses provided below address the thoughtful comments from the reviewers and significantly improve the quality of this manuscript.
Comments and Suggestions for Authors
The manuscript entitled “Deep Generative Models for the Discovery of Antiviral Peptides Targeting Dengue Virus: A Systematic Review” addresses a relevant issue: the absence of specific antiviral treatments for dengue, the therapeutic potential of antiviral peptides (AVPs), and the use of deep generative models (DGMs) in their discovery.
- We appreciate the reviewer’s thoughtful assessment and valuable input on our manuscript. Recognizing the importance of improving both the content and structure, we have undertaken a thorough revision to address the issues raised.
- Each comment has been carefully considered, and our detailed responses and corresponding changes are outlined below.
Section 1 is appropriate and sufficiently introduces the reader to the clinical, therapeutic, and computational context of the problem.
- We sincerely appreciate your positive comments and are grateful for your constructive feedback throughout the review process.
The manuscript includes numerous recent sources (2020–2024), including peer-reviewed articles and specialized databases. However, Section 2 (Methodology) would benefit from a more detailed description of the inclusion/exclusion criteria and potential biases, in order to strengthen methodological transparency and systematic rigor.
- Thank you for your comment. The Methodology section has been revised accordingly and is now addressed on page 2, line 87 of the updated manuscript.
Both the title and abstract indicate that the main focus of the review is the use of DGMs to design AVPs targeting the dengue virus. Within this framework, virological background information should be presented accurately but concisely, without dominating the manuscript or diverting from its central focus.
- Thank you for your valuable comment. We agree with the reviewer that the virological background information was overly extensive. Accordingly, we have condensed the virological background information in the manuscript as suggested, streamlining the background section (Section 3 Background) in line with each of the reviewer’s specific recommendations.
Sections 3.1 to 3.11, currently included under the Results section, do not reflect outcomes derived from the systematic search, but rather provide a general overview of virological aspects of DENV (epidemiology, serotypes, viral proteins, pathogenic mechanisms, among others). While this information is valuable as background, it is overly developed and creates an imbalance in the manuscript, which should prioritize AVPs and AI models. It is recommended that these contents be condensed, consolidating repeated ideas (e.g., the role of NS1 or serotype variability) into a more concise narrative (1–2 paragraphs per topic), ideally as part of the Background section. This would allow more space for a comparative analysis of generative models, bioinformatics tools, and experimental validation, which represent the core of the review.
- Based on the reviewer’s suggestions, we have made the following revisions accordingly:
- The following sections have been removed as they are not closely aligned with the core focus of the manuscript: “3.2. WHO classifications”, “3.8. Role of nonstructural protein 1 (NS1) viral antigen”, “3.9. DENV genome variation and subgenomic RNA”, “3.10. Antibody-dependent enhancement (ADE)”, and “3.11. Role of Host Genetic Factors”.
- The subsection "Epidemiology of Dengue" (now presented in Section 3.2, page 5) has been condensed for brevity and clarity.
- The content from “3.7. Risks and mechanisms correlated with Dengue severity” has been integrated into “3.1. Overview of Dengue virus (DENV)” (page 4) to enhance narrative continuity.
- The section “3.5. Dengue treatment” has been revised and renamed as “3.3. Dengue treatment and vaccination” (page 5) in the updated version.
- In line with comments from another reviewer, “3.6. Challenges in controlling Dengue infection” has been rewritten and retitled as “3.4. Drug development efforts targeting Dengue virus and related flaviviruses” (page 6) in the revised manuscript.
Sections 3.12 to 3.21 are aligned with the objectives of the manuscript and provide appropriate context. Since the topics discussed exceed my area of expertise, I will not comment on their relevance, content, or depth of development. I will focus solely on the writing aspects. Some points for revision include:
Section 3.12 does not provide relevant information and could be removed.
- Section 3.12 has been removed as per the reviewer’s suggestion.
Section 3.13 repeats concepts already introduced in the introduction and may be shortened. I suggest reducing it by eliminating lexical and conceptual redundancies. Additionally, Figure 6 mentioned in this section is unclear—it is not evident whether the terms listed on the left refer to chemical molecules and those on the right to antiviral peptides. The arrows add confusion.
- Regarding the content in Section 3.12, we have removed the parts that were redundant with the Introduction to avoid repetition and improve clarity.
- For Figure 6, we sincerely appreciate the reviewer’s comment highlighting the potential confusion for readers. In response, we have revised the figure by removing the unnecessary arrows. This figure has been updated and is now presented as Figure 4 in the revised version of the manuscript (page 7, line 246).
Section 3.14 also has limited specific relevance and could be summarized or integrated into another section.
- The key contents of this section have been integrated into the revised version, specifically in Section 3.6 (page 7, lines 248–251).
Sections 3.18 and 3.19, although addressing relevant computational techniques (data augmentation, encoding, K-mers, embeddings), exceed the depth required for a systematic review unless explicitly linked to AVPs applied to DENV.
- Thank you for this appropriate suggestion. Sections 3.18 and 3.19 have been removed in the revised version of the manuscript.
Moreover, in Section 4—particularly subsections 4.1 and 4.2—general definitions and benefits of AVPs and AI are repeated without advancing toward critical analysis. This limits the contribution for an informed reader. It is suggested to enhance the analytical component by comparatively discussing model limitations, their practical applications in the context of dengue, and unexplored opportunities.
- We appreciate the reviewer’s valuable suggestions and have revised the manuscript accordingly. Specifically, we have incorporated new content and reorganized the Discussion section for improved clarity and alignment with the journal’s format requirements.
- Section 4.1 has been revised and moved to Section 5.5 titled “Limitations and opportunities of DGMs in Dengue-specific AVP discovery” in the updated version of the manuscript (Page 16, Line 496), following the template modification request.
- Additionally, we have expanded the discussion of practical applications in Sections 5.3 and 5.4 (Page 14, Line 428 and Page 15, Line 489, respectively).
Section 4.5 (Future Perspectives) lacks specificity. It would be advisable to more clearly articulate how the findings reviewed could drive concrete developments, such as in silico or in vitro validations, preclinical testing, or clinical integration strategies for AI-generated AVPs.
- The section on Future Perspectives has been revised accordingly and can now be found in Section 5.7 (Page 17, line 536).
The critical discussion could be expanded to include reflections on biases and limitations of current models; comparison between reviewed studies; cross-validation; and reproducibility.
- As suggested, the relevant content has been integrated into Sections 5.3, 5.4, and 5.5.
Additionally, the tone remains more descriptive than analytical in parts of sections 4.1 and 4.2, where general definitions or already-discussed advantages of AVPs or AI are reiterated instead of addressing their comparative impact, real-world limitations, or direct applications to DENV.
- As suggested, the relevant content has been integrated into Sections 5.3, 5.4, and 5.5.
Section 5 (Conclusion) is clear and logically structured, though somewhat lengthy and includes repetitions from the abstract and discussion, with a focus on summarization. A more forward-looking reflection is recommended: for instance, what concrete challenges must be overcome to apply these models to AVPs against DENV? What future steps are advisable for experimental or clinical validation? What short- or medium-term impact could this technology have?
- The suggested revisions have been incorporated into Section 6, page 18, line 555.
Comments on style and writing:
In the abstract, Latin terms such as in silico, in vitro, in vivo should appear in italics. Abbreviations not used repeatedly in the abstract (e.g., VAEs, GANs) need not be defined there.
- The terms in silico, in vitro, and in vivo have been italicized as requested.
- Additionally, the abbreviations VAEs and GANs have been revised in the Abstract according to the recommendation.
Throughout the manuscript, terminological consistency (e.g., AVPs, DGMs, DENV) and uniform acronym usage should be ensured. Acronyms defined in the abstract or introduction should not be redefined or written out again (e.g., L61, L69, and L360/368/371/374/558/638/657/661). Abbreviations introduced in the main text should remain consistent across all sections, including headings, figures, captions, and tables.
- All abbreviations have been carefully reviewed and consistently used throughout the manuscript, including in section headings, figure labels, captions, and tables.
The use of unnecessary capital letters should be revised, particularly in section titles and common nouns.
- The suggested revisions have been carefully reviewed and incorporated throughout the manuscript.
From L52 onward, following editorial guidelines such as ICMJE, AMA, and common scientific writing standards:
Disease names are written in lowercase (e.g., dengue, malaria), unless at the beginning of a sentence.
Virus names, as taxonomic proper nouns, are capitalized: Dengue virus, Influenza A virus, etc.
Virus acronyms are written in uppercase (e.g., DENV, HIV, HBV) and, once established, should be used consistently throughout the manuscript. For instance, “Dengue virus” or “Dengue virus type 1” should not be repeated after “DENV” and “DENV-1” have been introduced.
- The suggested revisions have been carefully reviewed and incorporated throughout the manuscript.
Specific corrections:
L124–L127: DENV-5 has been acknowledged in the scientific community as a potential fifth dengue virus serotype but was not included by the ICTV among classified species and thus is not formally recognized as a DENV type. It was identified in 2007 from a patient sample in Malaysia and published in 2013. It differs from the other four serotypes by following a sylvatic cycle (transmission between wild animals and mosquitoes) rather than the human-mosquito-human transmission of DENV-1 to DENV-4. Source: https://ictv.global/report/chapter/flaviviridae/flaviviridae/orthoflavivirus
- The relevant information and a supporting reference have been added in Section 3.1, lines 134–137.
L153: Avoid redundancy. Suggested revision: “The most common DENV-1 genotype was genotype I, including strains isolated between 2015 and 2017.”
- Thank you for the suggestion. The sentence has been revised as follows “The most common genotype was DENV-1, including strains isolated between 2015 and 2017” (Section 3.2, lines 175–176).
L188: Update information: In May 2024, WHO prequalified a new dengue vaccine (TAK-003), adding to the existing CYD-TDV vaccine developed by Sanofi Pasteur. These vaccines and their use are detailed in reference 25 of the manuscript. Source: https://www.who.int/news/item/15-05-2024-who-prequalifies-new-dengue-vaccine
- The relevant information and corresponding references have been added to Section 3.3, lines 190–191.
L278: Standardize title formatting (capitalize or lowercase consistently across all section headings).
- The suggested revisions have been carefully reviewed and incorporated throughout the manuscript.
L461/471: Italicize K-mer and standardize capitalization in “(binary encoding)”.
- As suggested by the reviewer, the content related to K-mer analysis in Section 3.19 has been removed in the revised version of the manuscript.
The remaining corrections are highlighted in the revised manuscript file.
- All revisions have been highlighted in red in the revised manuscript.

Reviewer 2 Report
Comments and Suggestions for Authors
Thank you for the opportunity to review the manuscript entitled " Deep Generative Models for the Discovery of Antivirals Peptides Targeting Dengue Virus: A Systematic Review". While the subject matter is timely and highly relevant to the field of computational drug discovery, the current version of the manuscript presents several major issues that preclude its acceptance in its current form. Below, I outline my key concerns and recommendations for substancial revision.
- Misalignment between Title, Objectives and Content. The title indicates a systematic review focused on deep generative models (DGMs) applied to the discovery of antiviral peptides against the Dengue Virus. However, the manuscript includes an extensive internal review of the Dengue virus itself - its classification, pathogenesis, and molecular biology - which, while informative, detracts from the core focus of the article. These sections are unnecessarily detailed for a review centered on computational peptide design and should be a considerably reduced or removed.
- Lack of Methodological Transparency and Reproducibility
- The manuscript claims to follow a systematic review approach but lacks essential elements such as:
- -a clearly defined search strategy (databases, exact search strings, Boolean logic)
- A PRISMA flow diagram: the diagram indicates that no records were excluded during the screening, retrieval, or elegibility assessment stages. This is highly unusual in systematics reviews and may raise concerns regarding the selectivity and robustness of the inclusion criteria. Please clarify wether this reflects the actual process or if the exclusions were not properly documented. the autors mention 31 studies included in a prior version of the review, and 11 newly included studies, totaling 42 studies. However, ti is not explained whether any of the previous studies were excluded or updated, wich makes difficult to assess the comprehensiveness and currency of your review. The box labeled "Other methods" appears to serve no purpose and adds unnecessary complexity to the diagram. I suggest removing it for clarity. It is unclear whether the review process was conducted by multiple reviewers independently, how disagrements were resolved. Including this information in the methodology would greatly strengthen the transparency and reproducibility of your review.
- The authors included review articles in their core analysis, which is a methodological error in the context of a systematic review intended to synthesize original research findings. Review papers should only be used to provide background information, not to form the evidence base of the systematic review itself.
- the manuscript does not provide a structured and quantitative summary of the studies included. Key aspects such as the number of antiviral peptides generated, the deep generative model architecture used (e.g., VAEs, GANs), training dataset details, sequence characteristics of the peptides (length, motifs, physicochemical properties) and validation techniques (in silico, in vitro), are either superficially mentioned or entirely absent. Moreover, although the manuscript states the 42 studies were included in the review, the data presented in the tables 7 and 8 appear to originate from a single reference only. There is no indication of how the information from the remaining studies was integrated, analysed or compared.
- Substancial portions of the manuscript are devoted to basic explanations of deep learning and generative models, without linking them directly to practical peptide discovery outcomes. this create an imbalance between theoretical exposition and practical, data-driven insigths. The authors should focus on evaluating how DGMs gave actually been applied to antiviral peptide design, including limitations and future directions.
While the topic highly relevant, the manuscript does not currently meet the standards of a systematic review. It needs major methodological restructuring, clarification of the literature search process, and a focused analysis of primary studies involving DGMs for peptide generation. If the autors choose to revise, I strongly recommend adhering to PRISMA guidelines and focused exclusively on original research articles that have implemented DGMs in antiviral peptide design against Dengue or related viruses.
Author Response
Response to Reviewer 2
Dear Reviewers, we genuinely appreciate all the comments and suggestions, which allowed us to improve the quality of the manuscript significantly. Our responses to the reviewer’s comments are provided below in a point-by-point fashion. All changes to the text in the revised manuscript are made and marked up using the red color highlight. We hope that our revisions and responses provided below address the thoughtful comments from the reviewers and significantly improve the quality of this manuscript.
Comments and Suggestions for Authors
Thank you for the opportunity to review the manuscript entitled " Deep Generative Models for the Discovery of Antivirals Peptides Targeting Dengue Virus: A Systematic Review". While the subject matter is timely and highly relevant to the field of computational drug discovery, the current version of the manuscript presents several major issues that preclude its acceptance in its current form. Below, I outline my key concerns and recommendations for substancial revision.
- We sincerely thank the reviewer for taking the time to evaluate our manuscript and for providing constructive suggestions. We acknowledge the need for a comprehensive revision and have carefully addressed all the comments provided.
- A detailed point-by-point response to each suggestion is provided below.
Misalignment between Title, Objectives and Content. The title indicates a systematic review focused on deep generative models (DGMs) applied to the discovery of antiviral peptides against the Dengue Virus. However, the manuscript includes an extensive internal review of the Dengue virus itself - its classification, pathogenesis, and molecular biology - which, while informative, detracts from the core focus of the article. These sections are unnecessarily detailed for a review centered on computational peptide design and should be a considerably reduced or removed.
- Based on the reviewer’s suggestions, we have made the following revisions to virological overview, accordingly:
- The following sections have been removed as they are not closely aligned with the core focus of the manuscript: “3.2. WHO classifications”, “3.8. Role of nonstructural protein 1 (NS1) viral antigen”, “3.9. DENV genome variation and subgenomic RNA”, “3.10. Antibody-dependent enhancement (ADE)”, and “3.11. Role of Host Genetic Factors”.
- The subsection "Epidemiology of Dengue" (now presented in Section 3.2, page 5) has been condensed for brevity and clarity.
- The content from “3.7. Risks and mechanisms correlated with Dengue severity” has been integrated into “3.1. Overview of Dengue virus (DENV)” (page 4) to enhance narrative continuity.
- The section “3.5. Dengue treatment” has been revised and renamed as “3.3. Dengue treatment and vaccination” (page 5) in the updated version.
- In line with comments from another reviewer, “3.6. Challenges in controlling Dengue infection” has been rewritten and retitled as “3.4. Drug development efforts targeting Dengue virus and related flaviviruses” (page 6) in the revised manuscript.
Lack of Methodological Transparency and Reproducibility
The manuscript claims to follow a systematic review approach but lacks essential elements such as:
-a clearly defined search strategy (databases, exact search strings, Boolean logic)
- The Methodology section has been revised accordingly and is now presented on Page 2, line 87.
A PRISMA flow diagram: the diagram indicates that no records were excluded during the screening, retrieval, or elegibility assessment stages. This is highly unusual in systematics reviews and may raise concerns regarding the selectivity and robustness of the inclusion criteria. Please clarify wether this reflects the actual process or if the exclusions were not properly documented. the autors mention 31 studies included in a prior version of the review, and 11 newly included studies, totaling 42 studies. However, ti is not explained whether any of the previous studies were excluded or updated, wich makes difficult to assess the comprehensiveness and currency of your review.
The box labeled "Other methods" appears to serve no purpose and adds unnecessary complexity to the diagram. I suggest removing it for clarity. It is unclear whether the review process was conducted by multiple reviewers independently, how disagrements were resolved. Including this information in the methodology would greatly strengthen the transparency and reproducibility of your review.
- Regarding the PRISMA flow diagram, we acknowledge that there was an error in recording the number of excluded records. Specifically, four papers were excluded at the beginning of the screening process because they integrated DGM into the generation and discrimination of peptide secondary structures, peptide classification tasks, or peptide-based biomarker prediction for clinical diagnosis, which were outside the scope of our defined review. Therefore, in the next step, no additional records were excluded and we recorded “0 records excluded” at this stage. This resulted in an error in the flow diagram representation.
- In response to the reviewer’s suggestion, we also removed all review articles from the set of studies included for data extraction. Accordingly, the PRISMA flow diagram has been updated to reflect these changes.
- Additionally, the box labeled "Other methods" has been removed from the diagram to improve clarity and consistency.
- Finally, in cases of disagreement between the two authors during the screening process, we resolved them by adhering to the principle of only retaining original research studies that employed DGMs specifically for generative tasks. Studies where DGMs were used merely as components in classification tasks were excluded.
The authors included review articles in their core analysis, which is a methodological error in the context of a systematic review intended to synthesize original research findings. Review papers should only be used to provide background information, not to form the evidence base of the systematic review itself.
- We sincerely appreciate the reviewer’s thoughtful comment regarding this methodological issue. In response, all review articles have been excluded from the data extraction and synthesis processes. Only original research articles reporting empirical results on the use of deep generative models (DGMs) for peptide generation were retained as part of the evidence base.
- The methodology section has been revised accordingly.
The manuscript does not provide a structured and quantitative summary of the studies included. Key aspects such as the number of antiviral peptides generated, the deep generative model architecture used (e.g., VAEs, GANs), training dataset details, sequence characteristics of the peptides (length, motifs, physicochemical properties) and validation techniques (in silico, in vitro), are either superficially mentioned or entirely absent.
Moreover, although the manuscript states the 42 studies were included in the review, the data presented in the tables 7 and 8 appear to originate from a single reference only. There is no indication of how the information from the remaining studies was integrated, analysed or compared.
- We thank the reviewer for highlighting areas for improvement. Firstly, we have expanded the discussion on practical applications in Sections 5.3 and 5.4 (Page 14, Line 428, and Page 15, Line 489, respectively). These sections now include the additional content as suggested.
- Regarding Table 7 and Table 8, we have revised and renamed them as Table 5 (Page 11) and Table 6 (Page 12), respectively, and provided separate and specific references for each.
Substancial portions of the manuscript are devoted to basic explanations of deep learning and generative models, without linking them directly to practical peptide discovery outcomes. this create an imbalance between theoretical exposition and practical, data-driven insigths. The authors should focus on evaluating how DGMs gave actually been applied to antiviral peptide design, including limitations and future directions.
- We have revised the manuscript by streamlining, removing, and adjusting content that was not directly relevant to the core focus of generative models. In addition, we have expanded and clarified the sections discussing the limitations, opportunities, and future directions of DGMs. These updates are reflected in Sections 5.5 (page 16) and 5.6 (page 17).
While the topic highly relevant, the manuscript does not currently meet the standards of a systematic review. It needs major methodological restructuring, clarification of the literature search process, and a focused analysis of primary studies involving DGMs for peptide generation. If the autors choose to revise, I strongly recommend adhering to PRISMA guidelines and focused exclusively on original research articles that have implemented DGMs in antiviral peptide design against Dengue or related viruses.
- Thank you for your valuable feedback. We acknowledge the limitations in the original structure and have made major revisions accordingly. Specifically, we have:
- Refined the methodology section to clarify the literature search strategy and selection process.
- Narrowed the focus to original studies applying DGMs for antiviral peptide design against Dengue and related viruses.
- Integrated key elements of the PRISMA guidelines where applicable.
- We believe these changes substantially improve the rigor and focus of the manuscript.

Reviewer 3 Report
Comments and Suggestions for Authors
In this review, the authors are trying to introduce some deep generative models for the discovery of antiviral peptides targeting dengue virus. The manuscript should be comprehensively revised. Below are some suggestions.
1) Since this is a review, the template for research article (Introduction-Method-Results-Discussion) is not very suitable.
2) What interests readers the most is how to use deep generative models to discover antiviral peptides targeting dengue virus, which should be the most important topic of this review. In this manuscript, the authors listed several anti-dengue peptides reported in previous literatures. Several deep generative models and their principles based on which the models work are also described. However, are there anti-dengue peptides which are designed or optimized by using deep generative models? Several paragraphs or one section is needed to tell us the examples about how the deep generative models is utilized for the discovery of anti-dengue peptides, which is actually the most important part of this review.
3) Some parts of the manuscript are unnecessarily long. The main content of this review should be deep generative models and how the models is used to anti-dengue peptides. So, besides the sections of Introduction and Discussion, the main body of this review should consist of three parts: 1) deep generative models and their principles; 2) anti-dengue peptides plus other antiviral peptides and antiviral peptide databases; 3) the examples of utilization of deep generative models in design and optimization of anti-dengue peptides and/or other antiviral peptides. Other information which is not relevant to the topic of this review should be briefly introduce in the section of Introduction or deleted, such as WHO classification, Epidemiology of Dengue, Virological characteristics, Dengue treatment, Challenges in controlling Dengue infection, Risks and mechanisms correlated with Dengue severity, Role of nonstructural protein 1 (NS1) viral antigen, DENV genome variation and subgenomic RNA, Antibody-dependent enhancement (ADE), Role of Host Genetic Factors, Challenges and prospects in Dengue antiviral therapy, and so on.
4) Some references are not appropriate. For example, in Table 7 and in Table 8, the authors listed several antiviral peptides against DENV structural proteins and non-structural proteins. However, only one review article was cited. As this manuscript is also a review article, it is not appropriate to cite another review article for so many antiviral peptides. The authors should cite each original paper for each peptide in Table 7 and Table 8.
Author Response
Response to Reviewer 3
Dear Reviewers, we genuinely appreciate all the comments and suggestions, which allowed us to improve the quality of the manuscript significantly. Our responses to the reviewer’s comments are provided below in a point-by-point fashion. All changes to the text in the revised manuscript are made and marked up using the red color highlight. We hope that our revisions and responses provided below address the thoughtful comments from the reviewers and significantly improve the quality of this manuscript.
Comments and Suggestions for Authors
In this review, the authors are trying to introduce some deep generative models for the discovery of antiviral peptides targeting dengue virus. The manuscript should be comprehensively revised. Below are some suggestions.
- We sincerely thank the reviewer for taking the time to evaluate our manuscript and for providing constructive suggestions. We acknowledge the need for a comprehensive revision and have carefully addressed all the comments provided.
- A detailed point-by-point response to each suggestion is provided below.
1) Since this is a review, the template for research article (Introduction-Method-Results-Discussion) is not very suitable.
- We have restructured the manuscript using the Introduction–Methodology–Background–Findings–Discussion template, in the hope that this format will more effectively convey the full scope and content of the paper.
2) What interests readers the most is how to use deep generative models to discover antiviral peptides targeting dengue virus, which should be the most important topic of this review. In this manuscript, the authors listed several anti-dengue peptides reported in previous literatures. Several deep generative models and their principles based on which the models work are also described. However, are there anti-dengue peptides which are designed or optimized by using deep generative models? Several paragraphs or one section is needed to tell us the examples about how the deep generative models is utilized for the discovery of anti-dengue peptides, which is actually the most important part of this review.
- Thank you for pointing out this issue in the manuscript.
- To the best of our knowledge, there have been no reported applications of deep generative models (DGMs) specifically developed for targeting the dengue virus. Thus far, architectures such as GANs and VAEs have primarily been utilized for the generation of antimicrobial or antiviral peptides in broader contexts. Notably, one study employing a LeakGAN framework explored AVPs associated with the Flaviviridae family; however, it did not specifically address the dengue viruses.
- We acknowledge that this reflects a current limitation in the field, but we also consider it a promising opportunity for future research aimed at applying DGMs to the development of dengue-targeted AVPs.
- We have revised the manuscript to expand the discussion on this topic. Additional explanations regarding the applications and potential of DGMs in AVP discovery have been included in Sections 5.3 (page 14) and 5.4 (page 15) of the revised manuscript.
3) Some parts of the manuscript are unnecessarily long. The main content of this review should be deep generative models and how the models is used to anti-dengue peptides. So, besides the sections of Introduction and Discussion, the main body of this review should consist of three parts: 1) deep generative models and their principles; 2) anti-dengue peptides plus other antiviral peptides and antiviral peptide databases; 3) the examples of utilization of deep generative models in design and optimization of anti-dengue peptides and/or other antiviral peptides. Other information which is not relevant to the topic of this review should be briefly introduce in the section of Introduction or deleted, such as WHO classification, Epidemiology of Dengue, Virological characteristics, Dengue treatment, Challenges in controlling Dengue infection, Risks and mechanisms correlated with Dengue severity, Role of nonstructural protein 1 (NS1) viral antigen, DENV genome variation and subgenomic RNA, Antibody-dependent enhancement (ADE), Role of Host Genetic Factors, Challenges and prospects in Dengue antiviral therapy, and so on.
- The revised manuscript has fully incorporated the contents recommended by the reviewers, including: (1) deep generative models and their principles; 2) anti-dengue peptides plus other antiviral peptides and antiviral peptide databases; 3) the examples of utilization of deep generative models in design and optimization of anti-dengue peptides and/or other antiviral peptides.
- In addition, we acknowledge the imbalance present in the previous version of the manuscript. In the revised version, we have removed and condensed the sections as suggested. These adjustments are now reflected in Section 3 (Background) of the updated manuscript.
4) Some references are not appropriate. For example, in Table 7 and in Table 8, the authors listed several antiviral peptides against DENV structural proteins and non-structural proteins. However, only one review article was cited. As this manuscript is also a review article, it is not appropriate to cite another review article for so many antiviral peptides. The authors should cite each original paper for each peptide in Table 7 and Table 8.
- Thank you for your insightful comment. Regarding Tables 7 and 8, we have now provided separate and specific references for each. These have been revised and presented as Table 5 (Page 11) and Table 6 (Page 12), respectively, in the updated manuscript.

Reviewer 4 Report
Comments and Suggestions for Authors
This is a reasonably good paper that touches on an important topic. But there are still many questions that are not addressed in this paper.
1) There is no description of the family of dengue virus or its family member. This is important because many in the family have similar characteristics in terms of clinical manifestations and biological activities and they use very similar proteins. What kind of virus-- RNA/DNA virus? How different are the relatives?
https://pmc.ncbi.nlm.nih.gov/articles/PMC7089231/
2) Related to (1), there is no mention of development efforts for drugs against its relatives. This is also important because they are likely to be effective against DENV:
https://pubmed.ncbi.nlm.nih.gov/24001228/
3) NS1/2 are not the only proteins that affect DENV morbidity and virulence. Disorder at M/PrM and C affects the morbidity and mortality rates.
https://pubmed.ncbi.nlm.nih.gov/31698857/
https://pubmed.ncbi.nlm.nih.gov/27102744/
4) I am a bit confused about the targets and its mechanism. The exposed proteins such as E and possibly M/PrM are the targets then there is no need to know if the peptide can enter the host cell or vision. For the other viral proteins, you may need to consider whether the peptide can enter the host cell and virion. How is the AI models addressing this issue? Is it even addressing this issue? Did they calculate the hydrophobicity of the peptide?
5) Reproducibility and Reliability. The paper describes in length the theoretical framework of AI models for the DENV drug development. That is all great and fine. The important question is whether the models are reproducible and reliable. Are there actual experimental data showing that the peptide designed actually works. There is no mention of this in the paper
Author Response
Response to Reviewer 4
Dear Reviewers, we genuinely appreciate all the comments and suggestions, which allowed us to improve the quality of the manuscript significantly. Our responses to the reviewer’s comments are provided below in a point-by-point fashion. All changes to the text in the revised manuscript are made and marked up using the red color highlight. We hope that our revisions and responses provided below address the thoughtful comments from the reviewers and significantly improve the quality of this manuscript.
Comments and Suggestions for Authors
This is a reasonably good paper that touches on an important topic. But there are still many questions that are not addressed in this paper.
- We deeply appreciate the reviewer’s thorough assessment and thoughtful feedback on our manuscript. The comments have been invaluable in helping us improve the clarity and quality of the work.
- A detailed point-by-point response to each suggestion is provided below.
1) There is no description of the family of dengue virus or its family member. This is important because many in the family have similar characteristics in terms of clinical manifestations and biological activities and they use very similar proteins. What kind of virus-- RNA/DNA virus? How different are the relatives?
https://pmc.ncbi.nlm.nih.gov/articles/PMC7089231/
- Thank you for the insightful suggestion and the valuable reference. We have incorporated additional descriptions of the dengue virus and its related family members in Section 3.4, page 6 of the revised manuscript.
2) Related to (1), there is no mention of development efforts for drugs against its relatives. This is also important because they are likely to be effective against DENV:
https://pubmed.ncbi.nlm.nih.gov/24001228/
- We have incorporated the relevant drug development efforts into Section 3.4, page 6 of the revised manuscript. In addition, the valuable reference suggested has been included in the updated version. We sincerely thank the reviewer for sharing this information.
3) NS1/2 are not the only proteins that affect DENV morbidity and virulence. Disorder at M/PrM and C affects the morbidity and mortality rates.
https://pubmed.ncbi.nlm.nih.gov/31698857/
https://pubmed.ncbi.nlm.nih.gov/27102744/
- We sincerely appreciate the reviewer’s suggestion and the two valuable references provided. The relevant updates have been incorporated into Section 3.1 (page 5, line 158) accordingly.
4) I am a bit confused about the targets and its mechanism. The exposed proteins such as E and possibly M/PrM are the targets then there is no need to know if the peptide can enter the host cell or vision. For the other viral proteins, you may need to consider whether the peptide can enter the host cell and virion. How is the AI models addressing this issue? Is it even addressing this issue? Did they calculate the hydrophobicity of the peptide?
- We sincerely thank the reviewer for highlighting this interesting point. We would like to offer two responses to this comment:
- First, existing generative frameworks typically optimize for general antiviral features or physicochemical properties, rather than incorporating virus-specific mechanisms (e.g., the NS1 or E proteins of DENV). This represents a current limitation of DGM-based studies in general and highlights a potential direction for future research aimed at developing anti-Dengue peptides. These points have been addressed in Section 5.5, page 16 of the revised manuscript.
- Second, several studies have proposed selection criteria based on physicochemical properties for both input peptide data and validation strategies of sequences generated by DGMs. For instance, criteria such as net charge at pH 7.4 ≥ -1, GRAVY > -1, sequence length between 7 and 30, and secondary structure characteristics such as helix probability > 0.35 and strand probability < 0.3 have been utilized. The GRAVY index (Grand Average of Hydropathy), reflecting the hydrophobicity of peptides, is also commonly applied. These discussions are included in Section 5.3, page 14 of the revised manuscript.
5) Reproducibility and Reliability. The paper describes in length the theoretical framework of AI models for the DENV drug development. That is all great and fine. The important question is whether the models are reproducible and reliable. Are there actual experimental data showing that the peptide designed actually works. There is no mention of this in the paper
- Thank you for the reviewer’s insightful comment.
- Currently, the validation strategies for antiviral peptides generated by deep generative models (DGMs) typically include physicochemical property-based assessments, similarity and diversity comparisons between real and generated sequences, as well as in silico antiviral activity prediction using benchmarked computational tools or web servers. However, actual experimental validation data remains limited. Therefore, we were unable to include detailed experimental results in our manuscript.
- Moving forward, a well-structured in-silico-to-in-vitro pipeline is essential for bridging the gap between computational prediction and experimental verification. These points have been addressed in Section 5.3 (page 14), Section 5.5 (page 16), and the Conclusion (page 18) of the revised manuscript.

Round 2
Reviewer 1 Report
Comments and Suggestions for Authors
The topic of the manuscript remains relevant and timely, as it addresses both the lack of specific antiviral therapies for dengue and the potential of deep generative models (DGMs) to accelerate the discovery of antiviral peptides (AVPs). The decision to restructure the article into a narrative review, rather than maintaining a systematic approach, was appropriate and responds well to the methodological limitations previously noted.
The title has been adjusted to "Literature Review," and the lack of criteria for classification as a systematic review has been clarified. The description of the search strategy was expanded; however, some ambiguities remain regarding the inclusion/exclusion criteria and potential biases.
The general information on DENV has been condensed and moved to the “Background” section, which improves the clarity of the manuscript. The figures were redesigned, and the explanatory captions have been enhanced—this represents a notable improvement.
Although a greater number of databases and generative models were included, the discussion remains largely descriptive. A comparative analysis and critical reflection on practical limitations or experimental validations are still missing.
The future perspectives section (4.5) remains somewhat general, lacking concrete application proposals. It would be beneficial to include more specific projections toward experimental or clinical validation, or the practical integration of AI in AVP development. This section could be enriched by concrete suggestions, such as in silico/in vitro validations, preclinical trials, or integration strategies with public health programs.
While the conclusion section is now more concise, it still repeats some ideas introduced earlier. It would benefit from a clearer projection of future applications or specific technical challenges.
Regarding writing style and scientific formatting, inconsistencies in the use of capitalization and abbreviations persist. They were pointed out in the text.

Author Response
Response to Reviewer 1
The topic of the manuscript remains relevant and timely, as it addresses both the lack of specific antiviral therapies for dengue and the potential of deep generative models (DGMs) to accelerate the discovery of antiviral peptides (AVPs). The decision to restructure the article into a narrative review, rather than maintaining a systematic approach, was appropriate and responds well to the methodological limitations previously noted.
- We sincerely appreciate the reviewer’s positive comments on the restructuring of our manuscript. We have addressed the remaining suggestions point by point as outlined below.
- All new modifications have been highlighted in Blue to distinguish them from previous revisions.
The title has been adjusted to "Literature Review," and the lack of criteria for classification as a systematic review has been clarified. The description of the search strategy was expanded; however, some ambiguities remain regarding the inclusion/exclusion criteria and potential biases.
- We appreciate the reviewer’s feedback. In response, we have revised and clarified the inclusion and exclusion criteria in the Methods section to enhance transparency.
The general information on DENV has been condensed and moved to the “Background” section, which improves the clarity of the manuscript. The figures were redesigned, and the explanatory captions have been enhanced—this represents a notable improvement.
- We sincerely thank the reviewer for the positive feedback regarding the restructuring of the Background section. We appreciate the recognition of our efforts to enhance the clarity and overall quality of the manuscript.
Although a greater number of databases and generative models were included, the discussion remains largely descriptive. A comparative analysis and critical reflection on practical limitations or experimental validations are still missing.
- Thank you for your insightful comments. In response, we have made the following revisions to improve the manuscript:
- We have added quantitative evaluation metrics such as ICâ‚…â‚€, ECâ‚…â‚€, and others for existing anti-DENV peptides in Tables 5 and 6 to better illustrate the therapeutic potential of ADPs.
- In Section 5.5, we have expanded our discussion of the practical limitations associated with current generative models. Additionally, redundant content in this section has been removed to enhance clarity and improve the overall quality of the manuscript.
- Further details on GANs, including the number of generated peptides, validation approaches, and associated data, are presented in Section 5.3. Additionally, the limitations concerning the lack of application data for VAE-based models are mentioned in Section 5.4.
The future perspectives section (4.5) remains somewhat general, lacking concrete application proposals. It would be beneficial to include more specific projections toward experimental or clinical validation, or the practical integration of AI in AVP development. This section could be enriched by concrete suggestions, such as in silico/in vitro validations, preclinical trials, or integration strategies with public health programs.
- We appreciate the reviewer’s insightful comment. We have revised the future perspectives (5.7. section in the revised manuscript) to include specific future directions such as in silico/in vitro validations, preclinical evaluations, and the integration of AI-assisted peptide discovery into public health initiatives.
While the conclusion section is now more concise, it still repeats some ideas introduced earlier. It would benefit from a clearer projection of future applications or specific technical challenges.
- We appreciate the reviewer’s helpful suggestion.
- In response, we have revised the Conclusion section to reduce redundancy and have added a clearer outlook on potential applications and the remaining technical challenges related to AI-driven AVP development.
Regarding writing style and scientific formatting, inconsistencies in the use of capitalization and abbreviations persist. They were pointed out in the text.
- We thank the reviewer for pointing out the inconsistencies in capitalization and abbreviation usage.
- These issues have been carefully reviewed and corrected throughout the manuscript as indicated.

Reviewer 3 Report
Comments and Suggestions for Authors
The manuscript has improved after revision.
Author Response
Comments and Suggestions for Authors
1. The manuscript has improved after revision.
- We sincerely thank the reviewer for the kind and encouraging comment.
Reviewer 4 Report
Comments and Suggestions for Authors
improvements seen.
Author Response
Comments and Suggestions for Authors
1. improvements seen.
- We thank the reviewer for the encouraging comment.